

# Non-unitarity maximizing unraveling of open quantum dynamics

Ruben Daraban[1], Fabrizio Salas-Ramírez[1,2] and Johannes Schachenmayer[1*]

**1** CESQ/ISIS (UMR 7006), CNRS and Université de Strasbourg, 67000 Strasbourg, France
**2** Département de Physique de l'École Normale Supérieure,
ENS-Université PSL, 75005 Paris, France

* schachenmayer@unistra.fr

## Abstract

The dynamics of many-body quantum states in open systems is commonly numerically simulated by unraveling the density matrix into pure-state trajectories. In this work, we introduce a new unraveling strategy that can adaptively minimize the averaged entanglement in the trajectory states. This enables a more efficient classical representation of trajectories using matrix product decompositions. Our new approach is denoted non-unitarity maximizing unraveling (NUMU). It relies on the idea that adaptively maximizing the averaged non-unitarity of a set of Kraus operators leads to a more efficient trajectory entanglement destruction. Compared to other adaptive entanglement lowering algorithms, NUMU is computationally inexpensive. We demonstrate its utility in large-scale simulations with random quantum circuits. NUMU lowers runtimes in practical calculations, and it also provides new insight on the question of classical simulability of quantum dynamics. We show that for the quantum circuits considered here, unraveling methods are much less efficient than full matrix product density operator simulations, hinting to a still large potential for finding more advanced adaptive unraveling schemes.



# 1  Introduction

Straightforward simulation of general quantum dynamics on classical computers is prevented by the exponential growth of the Hilbert space size with the particle number, $2^n$ in the case of a quantum register with $n$ qubits. This however does not imply that specific quantum circuits can not be classically simulated in an efficient way, Clifford circuits being a famous example [1]. Beyond Clifford circuits, a method to simulate generic quantum evolution efficiently relies on systematically truncating the Hilbert space making use of the concept of matrix product states (MPS) [2–4], also known as "tensor trains" [5].

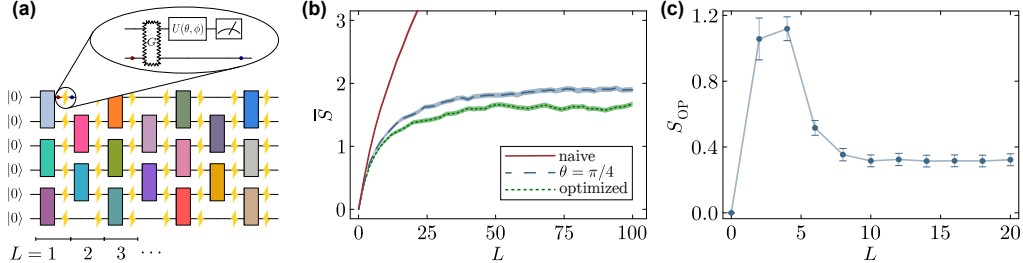

Figure 1: *Entanglement in unravelings of random circuits* – (a) Sketch of the quantum circuit setup used: Alternating layers of Haar random two-qubit gates are applied between nearest-neighbor qubits, inducing entanglement. These are followed by single-qubit noise channels that reduce entanglement. (Inset) The unitary freedom of the Kraus operators [Eq. (2)] corresponds to a choice of measurement basis after entanglement with an ancilla qubit, via a gate G [defined in Eq. (19)]. (b) Trajectory entanglement (TE) [defined in Eq. (8)] after each noise layer ($L$) for circuits with $n = 100$ qubits undergoing amplitude damping at rate $p_{\mathrm{ad}} = 0.22$. Average over $n_T = 250$ trajectories with bond dimension cutoff $\chi = 512$. The 3 lines correspond to different trajectory unraveling strategies. Solid red line: "naive unraveling" with Kraus operators $\hat{E}^{\mathrm{ad}}_{1,2}$ [defined in Eq. (16)]. Blue dashed line: "rotated" Kraus operators ($\theta = \pi/4$), $\hat{F}^{\mathrm{ad}}_{\pm} = (\hat{E}^{\mathrm{ad}}_1 \pm \hat{E}^{\mathrm{ad}}_2)/\sqrt{2}$. Green dotted line: adaptive NUMU unraveling [introduced in Sec. 3]. (c) Operator space entanglement (OE) evolution in a matrix product density operator simulation [defined in Eq. (35)] for the same model as in (b), with an averaging over 20 random circuit instances.

An MPS can be thought of as a generalization of a product state. While a product state only requires $2n$ real parameters to describe $n$ separable qubits, an MPS uses more parameters to represent quantum states with a controllable amount of entanglement. The number of parameters in an MPS depends on its "bond dimension" cutoff, $\chi$. Specifically, an MPS with bond dimension cutoff $\chi$ allows us to represent a pure quantum state $|\psi\rangle$ with finite bipartite entanglement entropy bounded by $\log_2 \chi$, i.e., $S_L \lesssim \log_2 \chi$. Here, $S_L$ is defined as the von Neumann entropy $S_L = -\sum_\alpha s_\alpha \log_2 s_\alpha$ with $s_\alpha$ the eigenvalues of the reduced density matrix $\hat{\rho}_L$ for a bipartition into two left/right blocks, $\hat{\rho}_L = \mathrm{tr}_R |\psi\rangle\langle\psi|$. Thus, the amount of entanglement plays a crucial role in determining how effectively an MPS can simulate a quantum many-body system [6, 7]. Various factors, such as disorder [8], the presence of homogeneous long-range interactions [9–11], and crucially for us, dissipation and decoherence [12–14], can significantly limit entanglement, making MPS a powerful tool for studying those effects.

However, how to take advantage of the presence of dissipation and decoherence to gain the best possible practical speedup in simulations remains an open question, since unlike in the case of pure states, the connection between entanglement and MPS simulability in large open quantum systems is much less understood [15]. This can be attributed to the fact that finding genuine quantum entanglement of large density matrices, as quantified for example by the entanglement of formation [16], is NP-hard [17, 18].

When relying on MPS methods, there are, aside from purification schemes [19], two main approaches to simulate the evolution of a large many-body density matrix $\hat{\rho}$: i) Either $\hat{\rho}$ is decomposed into a matrix product density operator form (MPDO) [20–23] or ii) $\hat{\rho}$ is unraveled into pure-state trajectories [24–28], and each trajectory state decomposed as an ordinary MPS [28]. The applicability of both approaches can be linked to bipartite "entanglement entropies". In the case of an MPDO representation, one can define a generalization of a pure-state bipartite entanglement entropy known as "operator (space) entanglement" (OE) [29–34]. For the quantum trajectory MPS (QT+MPS) scheme, one can compute the average bipartite entanglement entropy across the pure state trajectories to define a quantity known as "trajectory entanglement" (TE) [35]. It is important to note that, despite their names, neither are genuine measures of non-separability. However, both are directly connected to the efficiency of MPDO or QT+MPS simulations, respectively. For an identical density matrix evolution, the dynamics of the two quantities can vary significantly and which method is more adapted to which situation has been the subject of recent studies [35–39].

The dynamics of trajectory entanglement (TE) and operator entanglement (OE) can be effectively studied in generic Haar random quantum circuits, which lack any inherent symmetries. In this work we will consider a "brickwork structure" with interspersed noise layers as sketched in Fig. 1(a). Fig. 1(b/c) shows example results for the evolution of TE and OE for $n = 100$ qubits, subjected to amplitude damping noise (at rate $p_{\mathrm{ad}} = 0.22$). The different lines in Fig 1(b) correspond to various unraveling strategies for QT+MPS, that we investigate in this work. Notably, different unravelings can exhibit qualitatively very different TE growth behaviors. On the other hand, OE exhibits the characteristic "entanglement barrier" [33, 34, 40]: rapid growth followed by a fall and then a plateau.

Unlike in the case of OE, the variability of TE underscores the importance of selecting an unraveling strategy that minimizes entanglement and thus maximizes computational efficiency in QT+MPS. This freedom in unraveling the density matrix evolution into stochastically evolving pure states arises from the infinite number of ways to decompose a density matrix into a mixture of pure states. While all decompositions yield the correct statistical results due to linearity, entanglement entropies, being non-linear functions of the state, can differ significantly across unravelings. The flexibility in selecting the unraveling strategy was first explored in [41], where different approaches were compared for random Brownian circuits under dephasing, governed by a Lindblad master equation. Crucially, it was shown that even with a

fixed noise rate, different unravelings can lead to distinct phases of entanglement growth. Two adaptive algorithms were introduced to adjust the unraveling strategy for each stochastically evolving trajectory: one based on analytical insights and another utilizing a greedy numerical optimization approach that modifies the unraveling at each time-step [42]. Furthermore, in [43], the quantum evolution under generic noise channels was analyzed and a method to identify improved unravelings by mapping the problem to a spin model alongside a heuristic algorithm for adaptive unravelings minimizing the entanglement of individual qubits were introduced. We note that the growth of TE in our random quantum circuit of choice is connected to the recently emerging field of measurement-induced phase transitions [44–48]. In this field it is analyzed how varying the measurement rate can drive transitions between phases with different entanglement scaling behavior, in the post-selected trajectories. In contrast, our interest is in changing the entanglement scaling by varying the Kraus operator representation of a quantum operation, and Kraus operators do not necessarily correspond to the projective measurements commonly used in measurement-induced phase transition studies.

A fundamental issue with a direct optimization approach to lowering TE is the fact that it is computationally demanding [41, 42]. The reason for this is that computing entanglement entropies in an MPS generally relies on expensive calculations of singular value decompositions, typically scaling with the bond dimension as $\mathcal{O}(\chi^3)$. The latter scaling is comparable to those of usual MPS update algorithms, such as Time-Evolving Block Decimation (TEBD) [2], thus prohibiting an efficient utilization of direct entanglement optimization. Here we address the challenge of finding a more computationally efficient entanglement optimization method.

In this work, we introduce a new adaptive quantum trajectory unraveling method, which is based on maximizing a quantity we term trajectory non-unitarity. This new approach, that we denote non-unitarity maximizing unraveling (NUMU), is easy to implement compared to direct optimization. It relies on an efficient calculation of trajectory non-unitarities with a $\mathcal{O}(\chi^2)$ computational scaling, which ensures that the adaptive optimization step does not become the dominant cost in TEBD-style simulation schemes. Focusing on amplitude damping and phase flip noise channels, we demonstrate the effectiveness of NUMU using small two-qubit calculations, analytical insight, as well as large-scale simulations of the random quantum circuit setup depicted in Fig. 1(a). We show that NUMU offers an improvement over the best non-adaptive unraveling. Instead of entanglement entropies, we define an effective Schmidt rank as a more faithful quantity for simulability by MPS. We also not only focus on the mean values of effective Schmidt ranks but analyze its full distribution function. NUMU provides a constant reduction of the effective Schmidt ranks in our simulations, leading to a potential speed-up by a factor of roughly two. However, when analyzing critical noise rates ($p_c$) that determine the crossover to a QT-simulable regime, we observe that NUMU exhibits the same $p_c$ compared to the most efficient non-adaptive unraveling. Interestingly, we find however that the adaptively chosen optimization parameters in NUMU can give direct hints to the presence of an area-law to volume-law transition in the entanglement scaling. Lastly, we find that MPDO simulations of the identical circuits are still fundamentally more feasibly allowing an efficient simulation of the quantum circuit to arbitrary depths. The operator entanglement barrier is low enough, even in regimes where the trajectories in QT+MPS and NUMU exhibit a volume law scaling. This highlights the strong potential for finding more advanced unraveling strategies, involving, e.g., also more than a single qubit at a time. For such more advanced methods, trajectory non-unitarites may provide a promising new starting point.

This paper is organized as follows. In Sec. 2, we briefly summarize the fundamental concepts of quantum trajectories, entanglement, MPS, entanglement optimization and introduce our model of interest. In Sec. 3, we then introduce our measure for trajectory non-unitarity. There, we discuss the behavior of this quantity and introduce the NUMU algorithm. In Sec. 4, we then apply NUMU to a large random circuit model and analyze its capability for reducing

MPS simulation complexity, and how it compares to MPDO simulations. Lastly, we provide a conclusion and outlook in Sec. 5.

## 2 Background: Quantum trajectories, entanglement, and MPS

**Quantum trajectories:** When describing the discretized evolution of an open quantum system coupled to an environment, such as in a noisy quantum circuit, it is natural to use the quantum operation formalism [1]. It describes the change of the system density operator $\hat{\rho}$ as a map $\mathcal{E} : \hat{\rho} \to \hat{\rho}'$, with the environment traced out. In its most general form, this map can be described with the operator sum representation:

$$\mathcal{E}(\hat{\rho}) = \sum_{j=1}^{m} \hat{E}_j \hat{\rho} \hat{E}_j^{\dagger}. \tag{1}$$

The Kraus operators $\left\{\hat{E}_j\right\}_{1 \leq j \leq m}$ are (generally non-Hermitian) matrices satisfying $\sum_j \hat{E}_j^{\dagger} \hat{E}_j = \mathbb{1}$.

An important property of this formalism is that the Kraus operators are not a unique representation of the map $\mathcal{E}$. In fact, any arbitrary $m' \times m$ matrix $U$ satisfying $U^{\dagger}U = \mathbb{1}$ can be used to find an alternative operator sum representation $\left\{\hat{F}_j\right\}_{1 \leq j \leq m'}$ for the same channel $\mathcal{E}$:

$$\hat{F}_j = \sum_{k=1}^{m} u_{jk} \hat{E}_k, \qquad \text{or} \qquad \begin{bmatrix} \hat{F}_1 \\ \vdots \\ \hat{F}_{m'} \end{bmatrix} = U \begin{bmatrix} \hat{E}_1 \\ \vdots \\ \hat{E}_m \end{bmatrix}. \tag{2}$$

Using this operator sum representation, the new open system density matrix $\mathcal{E}(\hat{\rho})$ can be interpreted as describing a stochastic process:

$$\mathcal{E}(\hat{\rho}) = \sum_{j=1}^{m} \text{tr}\left(\hat{F}_j \hat{\rho} \hat{F}_j^{\dagger}\right) \frac{\hat{F}_j \hat{\rho} \hat{F}_j^{\dagger}}{\text{tr}\left(\hat{F}_j \hat{\rho} \hat{F}_j^{\dagger}\right)} = \sum_{j=1}^{m} p_j \hat{\rho}_j, \tag{3}$$

with probabilities $p_j = \text{tr}\left(\hat{F}_j \hat{\rho} \hat{F}_j^{\dagger}\right)$ and possible outcomes $\hat{\rho}_j = \hat{F}_j \hat{\rho} \hat{F}_j^{\dagger}/p_j$. The non-determinism originates from quantum measurements: each outcome for the system state $\hat{\rho}_j$ corresponds to a partial projective measurement performed only on the environment subspace (see inset in Fig 1a). Then, since the outcome of those measurements is unknown, considering all outcomes amounts to tracing out the environment degrees of freedom. The different outcome states $\hat{\rho}_j$ depend on the choice of the measurement basis. The choice of this basis is directly linked to the unitary degree of freedom in the Kraus operators choice in Eq. (2).

This interpretation gives rise to the quantum trajectory (QT) method. In QT, the density matrix is approximated by averaging over $n_T$ pure-state trajectories:

$$\hat{\rho} \approx \frac{1}{n_T} \sum_{k=1}^{n_T} \left|\psi^{(k)}\right\rangle\left\langle\psi^{(k)}\right|. \tag{4}$$

For a general map defined in Eq. (3), the unraveling $\left\{\left|\psi^{(k)}\right\rangle\right\}_{1 \leq k \leq n_T}$ of the full density matrix $\hat{\rho}$ can be updated with a non-deterministic scheme, $\left|\psi^{(k)}\right\rangle \xrightarrow{\mathcal{E}} \left|\psi_u^{(k)}\right\rangle$:

$$\left|\psi_u^{(k)}\right\rangle = \begin{cases} \hat{F}_1 \left|\psi^{(k)}\right\rangle / \sqrt{p_1}, & \text{with probability} \quad p_1 = \left\langle\psi^{(k)}\right|\hat{F}_1^{\dagger}\hat{F}_1\left|\psi^{(k)}\right\rangle, \\ \quad\vdots \\ \hat{F}_m \left|\psi^{(k)}\right\rangle / \sqrt{p_m}, & \text{with probability} \quad p_m = \left\langle\psi^{(k)}\right|\hat{F}_m^{\dagger}\hat{F}_m\left|\psi^{(k)}\right\rangle. \end{cases} \tag{5}$$

Using Eq. (3), it is clear that $\hat{\rho}' = \mathcal{E}(\hat{\rho}) \approx \sum_{k}^{n_T} |\psi_u^{(k)}\rangle\langle\psi_u^{(k)}| / n_T$ for $n_T \gg 1$. Unraveling the discretized dynamics of open quantum systems provides an efficient way to simulate density matrices for large systems. In this context QT has proven to be one of the most effective methods for numerically simulating quantum dynamics, e.g., when discretizing Lindblad master equations [24–28]. This approach is particularly advantageous when QT is integrated with matrix product state (MPS) representations.

**Entanglement, Schmidt decomposition and MPS:** We briefly review the definitions of entanglement entropies and their connection to representability using MPS. Our focus lies on the entanglement evolution in individual trajectories, $\left\{|\psi^{(k)}\rangle\right\}_{1 \leq k \leq n_T}$, specifically examining the entanglement between bipartitions of the pure states. It is important to emphasize that this measure does not reflect genuine entanglement in the density matrix evolution. However, it remains a useful metric, especially for assessing MPS simulation efficiency. A convenient starting point for describing the entanglement properties of a pure state is the Schmidt decomposition. For a bipartition of a system into subsystems $A$ and its complement $B$, the Schmidt decomposition rewrites the state using orthogonal basis vectors of $A$ and $B$:

$$\left|\psi^{(k)}\right\rangle = \sum_{\alpha=1}^{\chi_k} \lambda_\alpha \left|\phi_\alpha^A\right\rangle \otimes \left|\phi_\alpha^B\right\rangle , \tag{6}$$

with $\chi_k$ being the Schmidt rank of the trajectory along the bipartition, $\lambda_\alpha$ the real Schmidt values, and $\langle\phi_\alpha^{A,B}|\phi_\beta^{A,B}\rangle = \delta_{\alpha\beta}$. Note that the states $\left|\phi_\alpha^{A,B}\right\rangle$ are the eigenvectors of the respective reduced density matrices $\hat{\rho}_{A,B} = \mathrm{tr}_{B,A}(\hat{\rho}_{AB})$ with eigenvalues $p_\alpha \equiv \lambda_\alpha^2$.

The von Neumann entanglement entropy quantifies the entanglement between the subsystems in one positive number. It is defined as the Shannon entropy of the probabilities given by the eigenvalues of the reduced density operators:

$$S\left(\left|\psi^{(k)}\right\rangle\right) = S\left(\hat{\rho}_A^{(k)}\right) = -\mathrm{tr}\,\hat{\rho}_A^{(k)} \log_2 \hat{\rho}_A^{(k)} = -\sum_{\alpha=1}^{\chi_k} \lambda_\alpha^2 \log_2 \lambda_\alpha^2 . \tag{7}$$

Instead of the von Neumann entanglement entropy, the Rényi entropies $S_n \equiv \frac{1}{1-n} \log_2\left(\mathrm{tr}\left(\hat{\rho}_A^n\right)\right)$ are also commonly used.

Based on the single state entropy, we define the trajectory averaged entanglement entropy ("trajectory entanglement", TE) by averaging over the pure state trajectories that unravel the open system dynamics:

$$\overline{S} \equiv \frac{1}{n_T} \sum_{k=1}^{n_T} S\left(\left|\psi^{(k)}\right\rangle\right) . \tag{8}$$

TE is a quantity of fundamental interest, and has been used for determining measurement-induced phase transitions by analyzing the scaling behavior of $\overline{S}$ as a function of space and time [44–48]. TE as defined in Eq. (8) is connected to the efficiency of QT simulations when using an MPS as state representation [35, 41, 42] (see below), however it is not unique and it depends on the unitary transformation used in Eq. (2).

A proper definition of entanglement for density matrices is given by the entanglement of formation (EoF) [16], which is defined as the minimum value of averaged entropy over all pure states ensembles $\left\{|\psi_j\rangle\right\}$ representing the density matrix $\hat{\rho}$:

$$E_f(\hat{\rho}) \equiv \min \sum_j p_j S\left(|\psi_j\rangle\langle\psi_j|\right) . \tag{9}$$

The EoF constitutes a lower bound for TE and our goal is to approach it as much as possible.

Matrix product states (MPS) can provide an efficient representation for states with limited entanglement entropies [2,4,6]. Using the $\Gamma-\lambda$ notation from [2], we can write the amplitudes of a trajectory state $\left|\psi^{(k)}\right\rangle = \sum_{i_1,i_2,\ldots i_n} c^{(k)}_{i_1,i_2,\ldots,i_n} |i_1\rangle \otimes |i_2\rangle \otimes \cdots \otimes |i_n\rangle$ as the tensor contraction:

$$c^{(k)}_{i_1,i_2,\ldots,i_n} = \sum_{\alpha_1}^{\chi^{[1]}} \cdots \sum_{\alpha_{n-1}}^{\chi^{[n-1]}} \Gamma^{[1]i_1}_{\alpha_1} \lambda^{[1]}_{\alpha_1} \Gamma^{[2]i_2}_{\alpha_1,\alpha_2} \lambda^{[2]}_{\alpha_2} \ldots \lambda^{[n-1]}_{\alpha_{n-1}} \Gamma^{[n]i_n}_{\alpha_{n-1}}. \tag{10}$$

In this full canonical MPS form, the trajectory state can be expressed locally, using the rank 3 (except at boundaries) tensors $\Gamma^{[\nu]i_\nu}_{\alpha_{\nu-1},\alpha_\nu}$ and the Schmidt coefficients $\lambda^{[\nu]}_{\alpha_\nu}$. For example, given a decomposition for a "bond" $\nu$, i.e., for the splitting of the qubit register into two blocks $[1\ldots\nu]$ and $[\nu+1\ldots n]$, the Schmidt decomposition is

$$\left|\psi^{(k)}\right\rangle = \sum_{\alpha_\nu}^{\chi^{[\nu]}} \lambda^{[\nu]}_{\alpha_\nu} \left|\phi^{L,\nu}_{\alpha_\nu}\right\rangle \otimes \left|\phi^{R,\nu}_{\alpha_\nu}\right\rangle, \tag{11}$$

where $\left|\phi^{L/R,\nu}_{\alpha}\right\rangle$ denotes the Schmidt bases for the blocks left/right block, respectively. Furthermore, the state can be expressed in the local basis $|i_\nu\rangle$ of qubit $\nu$, using

$$\left|\psi^{(k)}\right\rangle = \sum_{i_\nu} \sum_{\alpha_{\nu-1}}^{\chi^{[\nu-1]}} \sum_{\alpha_\nu}^{\chi^{[\nu]}} \lambda^{[\nu-1]}_{\alpha_{\nu-1}} \Gamma^{[\nu]i_\nu}_{\alpha_{\nu-1},\alpha_\nu} \lambda^{[\nu]}_{\alpha_\nu} \left|\phi^{L,\nu-1}_{\alpha_{\nu-1}}\right\rangle \otimes |i_\nu\rangle \otimes \left|\phi^{R,\nu}_{\alpha_\nu}\right\rangle. \tag{12}$$

Central to MPS methods is the ability to construct approximate representations of quantum states by restricting the Schmidt ranks $\chi^{[\nu]}$, or "bond dimensions", across the $n-1$ bonds to a restricted cutoff value of $\chi$, i.e., such that after some truncation $\chi^{[\nu]} \leq \chi$. For bond $\nu$, this approximation is equivalent to truncating the probability distribution $p^{[\nu]}_\alpha = \left(\lambda^{[\nu]}_\alpha\right)^2$ below a certain threshold. Thus, at bond $\nu$ for a certain tolerance $\epsilon > 0$ there will be a cutoff bond dimension $\chi$ required to keep the sum of the discarded probabilities, also known as truncation error, below this tolerance: $p^{[\nu]}_{\text{tr}} \equiv \sum_{\alpha>\chi} p^{[\nu]}_\alpha < \epsilon$.

Instead of entanglement entropies, as a more practical figure of merit for MPS state representability, we suggest to focus on an upper bound for the cutoff. For a tolerance $\epsilon$, we denote this quantity as *effective Schmidt rank*, $\chi^{[\nu],\epsilon}_{\text{eff},k}$ for trajectory $k$ and bond $\nu$. As pointed out in [49], such an upper bound can be computed from the "mean index" and its standard deviation:

$$\mu^{[\nu]}_k \equiv \sum_{\alpha=1}^{\chi_k} \alpha\, p^{[\nu]}_\alpha, \qquad \sigma^{[\nu]}_k \equiv \sqrt{\sum_{\alpha=1}^{\chi_k} p^{[\nu]}_\alpha \alpha^2 - \left(\mu^{[\nu]}_k\right)^2}, \tag{13}$$

where we assume the probabilities $p_\alpha$ sorted in descending order. The rigorous upper bound follows from a simple argument involving Chebyshev's inequality (see Appendix. A). When defining

$$\chi^{[\nu],\epsilon}_{\text{eff},k} \equiv \mu^{[\nu]}_k + \frac{\sigma^{[\nu]}_k}{\sqrt{\epsilon}}, \tag{14}$$

as long as the cutoff is chosen larger or equal than the effective Schmidt rank, $\chi \geq \chi^{[\nu],\epsilon}_{\text{eff},k}$, then for the given bond $\nu$ (in trajectory $k$), the truncation error is guaranteed to be bounded by $p^{[\nu]}_{\text{tr}} \leq \epsilon$. It is important to note that in practice this implies that the true truncation error can be much smaller than $\epsilon$. We find a choice of $\epsilon = 10^{-4}$ generally suitable as in practice it leads to values where $\chi_{\text{eff}}$ is of similar magnitude as the actually chosen cutoff $\chi$.

The simple rigorous guarantee for representability within a given error $\epsilon$ is an advantage of the effective Schmidt ranks compared to entanglement entropies, which depend on the choice of entropy measure. For example, as shown in [6], the scaling behavior of Rényi entanglement entropies $S_n$ with order $n \geq 1$ can indicate inapproximability, but can not indicate efficient approximability. Therefore, when using $S_{n \geq 1}$, and in particular the von Neumann entanglement entropy, a transition from a volume-law entanglement phase to an area-law entanglement phase is not enough to indicate efficient approximability by an MPS, and the scaling of a Rényi entropy with $n < 1$ is needed. In particular, in the context of measurement-induced phase transitions it has been shown that the critical rate $p_c$ can depend on the Rényi index $n$ [46]. Using the minimum bond dimension $\chi_{\text{eff},k}^{[\nu],\epsilon}$ avoids such issues.

In the theoretical thermodynamic limit, for a trajectory state written in a 1D MPS representation [as in Eq. (10)] to faithfully describe the quantum state, it is necessary that $\chi_{\text{eff},k}^{[\nu],\epsilon}$ (for $\epsilon \to 0$) remains constant as a function of the size of the qubit block $[1 \dots \nu]$. This corresponds to a scenario where the quantum state obeys an area law, for $S_{n < 1}$. Similarly, when simulating a layered quantum circuit to arbitrary depth, or the time evolution in a many-body system to arbitrary long times, $\chi_{\text{eff},k}^{[\nu],\epsilon}$ should remain bounded as a function of the circuit layer $L$ or time, respectively. Although a polynomial increase in $\chi_{\text{eff},k}^{[\nu],\epsilon}$ (or a logarithmic increase in $S_{n < 1}$) can still be considered computationally efficient, in practice it imposes a limit on the simulation depth. Thus, the statistics of $\chi_{\text{eff},k}^{[\nu],\epsilon}$ over trajectories can indicate whether the system is in a phase where it can be classically simulated with QT+MPS, or not. We denote this as QT-simulable or QT-non-simulable, respectively. Effective Schmidt ranks furthermore give an idea of the runtime of MPS algorithms due to the $\mathcal{O}(\chi^3)$ cost of common update schemes, such as the TEBD algorithm for applying gates [2].

As a main figure of merit, we define the trajectory averaged effective Schmidt rank for a tolerance $\epsilon > 0$ as

$$\chi_{\text{eff}}^{\epsilon} = \frac{1}{n_T} \sum_{k=1}^{n_T} \chi_{\text{eff},k}^{[\nu],\epsilon}. \tag{15}$$

Note that the averaged effective Schmidt rank $\chi_{\text{eff}}^{\epsilon}$ is here defined for a specific bond $\nu$, but in practical simulations, we will also perform an averaging over several bonds in the center of the system. As mentioned above we find it convenient to choose $\epsilon = 10^{-4}$, but highlight again that truncation errors can be much smaller than $10^{-4}$. For this choice $\chi_{\text{eff}}^{\epsilon}$ is not necessarily connected to the actual bond dimension $\chi$ used in the MPS, but we find both quantities to be of similar size. In large system simulations, as for entanglement entropies, convergence checks with $\chi$ are indispensable for extracting values of $\chi_{\text{eff}}^{\epsilon}$. We indeed also find that $\chi_{\text{eff}}^{\epsilon}$ is a stricter measure for representability, as we observe that it typically requires larger cutoff bond dimensions $\chi$ for convergence than the von Neumann entanglement entropy (see Appendix E).

**Random circuit model:** In this work, we consider quantum circuits that consist of random two qubits gates sampled from the unitarily invariant Haar measure arranged in a brickwork architecture and interspersed with single qubit Kraus channels after each layer. This setup is sketched in Fig. 1(a). Investigating such random circuits is worthwhile for understanding entanglement dynamics since in the absence of conservation laws or structure beyond locality, this is one of the most generic setups that leads to linear entanglement growth. The local structure of these random circuits, dictated by the brickwork arrangement of two-qubit gates, results in a light-cone expansion where information spreads outward by one qubit with each successive layer.

Previous work in the context of measurement-induced phase transitions established that when considering random circuits interspersed with projective measurements occurring at a certain rate $p$, there exists a critical measurement rate $p_c$ that separates two phases with different TE growth [48]. For $p < p_c$, TE grows linearly and becomes extensive, eventually saturating at a value proportional to the subsystem size (volume-law phase). For $p > p_c$, TE is suppressed, preventing it from becoming extensive (area-law phase). In the model considered here, projective measurements correspond to the case where the Kraus operators, $\{E_j\}_{1 \leq j \leq m}$, are both Hermitian, $\hat{E}_j = \hat{E}_j^\dagger$, and projective, $\hat{E}_j^2 = \hat{E}_j$. However, for most quantum operations this is not the case, and the Kraus operators correspond to a generalized measurement instead. This generalized measurement can be interpreted as the result of entanglement with ancillary qubits, which are then subject to projective measurements [1]. The unitary degree of freedom in the Kraus operator representation corresponds to the choice of an arbitrary measurement basis for the ancilla qubit [as sketched in Fig. 1(a)].

As noise types, here we consider paradigmatic single qubit amplitude damping and phase flip channels, each described by two Kraus operators. For amplitude damping at rate $p_{\mathrm{ad}}$ we use:

$$\hat{E}_1^{\mathrm{ad}} = |0\rangle\langle 0| + \sqrt{1 - p_{\mathrm{ad}}}|1\rangle\langle 1|, \quad \text{and} \quad \hat{E}_2^{\mathrm{ad}} = \sqrt{p_{\mathrm{ad}}}|0\rangle\langle 1|. \tag{16}$$

It corresponds to spontaneous decay from $|1\rangle$ to $|0\rangle$. It does not have the property of mapping the maximally mixed state to itself (it is non-unital) and its representation cannot be written in a form such that both Kraus operators are either Hermitian or unitary (up to a multiplicative constant). For phase flip, or "dephasing noise", at rate $p_{\mathrm{pf}}$ we use

$$\hat{E}_1^{\mathrm{pf}} = \sqrt{1 - p_{\mathrm{pf}}}\mathbb{1}, \quad \text{and} \quad \hat{E}_2^{\mathrm{pf}} = \sqrt{p_{\mathrm{pf}}}\hat{\sigma}_z, \tag{17}$$

with $\mathbb{1}$ the $2 \times 2$ identity and $\hat{\sigma}_z$ the Pauli z matrix. This channel is unital and the Kraus operators are unitary and Hermitian (up to the respective factors). For the random circuits we consider, results for bit flip ($\hat{\sigma}_x$ instead of $\hat{\sigma}_z$) or bit-phase flip ($\hat{\sigma}_y$ instead of $\hat{\sigma}_z$) channels would lead to similar conclusions, since the preferred basis of the initial state ($|00\ldots0\rangle$ in the z basis here) becomes irrelevant after a few circuit layers.

**Entanglement optimization:** Our main objective is to utilize the unitary freedom of the Kraus operators from Eq. (2) to lower the entanglement in the pure state quantum trajectories generated by stochastic update procedure described in Eq. (5). Since we focus single on qubit noise channels, each with only 2 Kraus operators, the transformations $U$ from Eq. (2) can be easily parametrized by only two angles $\theta$ and $\phi$ [see Appendix. B for a more detailed discussion]:

$$U(\theta, \phi) = \begin{pmatrix} \cos\theta & \sin\theta \\ -\sin\theta & \cos\theta \end{pmatrix}\begin{pmatrix} e^{i\phi} & 0 \\ 0 & e^{-i\phi} \end{pmatrix}, \tag{18}$$

Formally, the unitary transformation of Kraus operators is equivalent to a unitary rotation by $U^\dagger$ on an ancillary qubit entangled via the gate

$$G = \hat{E}_1 \otimes \mathbb{1} + \hat{E}_2 \otimes \hat{\sigma}_x. \tag{19}$$

The choice of $\theta$ and $\phi$ at each stochastic update according to Eq. (5) does not change the overall density matrix evolution but can strongly influence the post-channel entanglement in the respective trajectory. At each channel application there is a choice of $\theta$ and $\phi$ leading to the lowest possible entanglement in the trajectory. Importantly, this does not necessarily result in the EoF for the full density matrix. Nevertheless, here we pursue this "greedy" approach, adaptively choosing $\theta$ and $\phi$ locally optimal for each qubit and trajectory separately [41, 42].

A straightforward numerical approach to find the best local choice of $\theta$ and $\phi$ directly is using the post-channel entanglement entropy as cost function. When using the entanglement entropy, this implies a search for the unitary transformation $U(\theta, \phi)$ such that

$$\mathcal{S}_{\text{pc}} = p_1 S(\hat{F}_1^{[\nu]} |\psi\rangle / \sqrt{p_1}) + p_2 S(\hat{F}_2^{[\nu]} |\psi\rangle / \sqrt{p_2}), \tag{20}$$

is minimized, where $p_j = \langle\psi|\hat{F}_j^\dagger \hat{F}_j|\psi\rangle$ and $\hat{F}_{1,2}(\theta, \phi)$ acts on the $\nu$-th qubit. Both depend on $\theta$ and $\phi$. This is known as greedy entanglement optimization and, following [42], we denote it by 2-GEO.

This method is generally expensive, as it requires computing the full Schmidt decomposition of the states $\hat{F}_{1,2}^{[\nu]} |\psi\rangle / \sqrt{p_{1,2}}$ as function of $\theta$, $\phi$. For an MPS with bond dimension $\chi$ the latter step requires a complexity of $\mathcal{O}(\chi^3)$. It can be performed as follows: Start by computing the $\mathcal{O}(\chi) \times 2 \times \mathcal{O}(\chi)$ tensors

$$A_{\alpha,i,\beta}^{(1,2)} = \sum_j f_{ij}^{(1,2)} \lambda_\alpha^{[\nu-1]} \Gamma_{\alpha,\beta}^{[\nu]j} \lambda_\beta^{[\nu]}, \tag{21}$$

where we used the local MPS expansion from Eq. (12) and the matrix elements of the Kraus operators $f_{ij}^{(1,2)} = (\hat{F}_{1,2})_{ij}$. Then, we renormalize the tensors $\tilde{A}_{\alpha,i,\beta}^{(1,2)} = A_{\alpha,i,\beta}^{(1,2)} / \sqrt{p_{1,2}}$ with $p_{1,2} = \sum_{\alpha,i,\beta} |A_{\alpha,i,\beta}^{(1,2)}|^2$. To obtain the new Schmidt values for a splitting of the state to the left or right of qubit $\nu$, perform a singular value decomposition of the $\mathcal{O}(\chi) \times \mathcal{O}(\chi)$ matrices $\tilde{A}_{\alpha,(i,\beta)}$ or $\tilde{A}_{(\alpha,i),\beta}$. The latter is the numerical bottleneck, since it leads to a complexity of $\mathcal{O}(\chi^3)$ and is thus as expensive as a usual update step with a two-qubit gate [2].

Due to the need for multiple computations of the Schmidt values in each optimization step, the 2-GEO approach is inefficient for large $\chi$ [41, 42]. Instead, it can be more useful to use analytical insight to find computationally inexpensive ways to lower the post-channel TE. In [41], an algorithm based on explicitly computing time derivatives of the TE for a Lindblad master equation with dephasing (corresponding to the phase flip channel) was introduced. In [43], a mapping to an exactly solvable spin model was used to determine the best non-adaptive choice of parametrization. There, an adaptive heuristic algorithm based on minimizing only single-qubit entanglement was also suggested. In the following section, we now define a new approach to an adaptive optimization algorithm, not based on entanglement minimization, but on a "Non-unitarity maximization".

## 3 Trajectory non-unitarity

Instead of using the post-channel entanglement entropy, we focus on another quantity that we denote *trajectory non-unitarity*. In this section, we define this quantity and use it to develop an adaptive non-unitarity maximizing unraveling (NUMU) algorithm, an efficient alternative 2-GEO. We discuss: i) how it compares to the EoF in the case of two qubits, and when using analytical insight; and ii) how it compares to 2-GEO.

Local (single-qubit) unitary operators leave the entanglement of a state invariant. Therefore, we argue that by maximizing the non-unitarity, entanglement is destroyed efficiently. Non-unitarity is generally a distance measure of how far a quantum channel is from being unitary, typically using an average over Haar random initial states [50–52]. Our new *trajectory non-unitarity* depends explicitly on the specific input state before applying the channel. In particular, we define the normalized output state after applying a non-unitary operator $\hat{O}$ as

$$|\psi_{\text{out}}\rangle = \frac{\hat{O}|\psi_{\text{in}}\rangle}{\sqrt{\langle\psi_{\text{in}}|\hat{O}^\dagger\hat{O}|\psi_{\text{in}}\rangle}} \equiv \hat{Q}(|\psi_{\text{in}}\rangle)) |\psi_{\text{in}}\rangle, \tag{22}$$

where we defined $\hat{Q}$ as a state dependent operator that preserves the normalization of $|\psi_{\text{in}}\rangle$. We can now define non-unitarity using the trace distance of $\hat{Q}^\dagger \hat{Q}$ from the identity $\mathbb{1}$, i.e., from

$$T\left(\hat{Q}^\dagger Q, \mathbb{1}\right) = \sqrt{\text{tr}\left[\left(\hat{Q}^\dagger \hat{Q} - \mathbb{1}\right)^\dagger \left(\hat{Q}^\dagger \hat{Q} - \mathbb{1}\right)\right]}. \tag{23}$$

To avoid computing the monotonic square root function, we now simply define

$$\mathcal{N}_{\mathcal{U}}\left(\hat{O}, |\psi_{\text{in}}\rangle\right) \equiv T\left(\hat{Q}^\dagger Q, \mathbb{1}\right)^2 = \text{tr}\left[\left(\hat{Q}^\dagger \hat{Q} - \mathbb{1}\right)^\dagger \left(\hat{Q}^\dagger \hat{Q} - \mathbb{1}\right)\right]. \tag{24}$$

Our goal is to maximize $\mathcal{N}_{\mathcal{U}}$ on average, with respect to the different unitary transformations $U$ from Eq. (2). Therefore, for the set of Kraus operators $\{\hat{F}_j\}_{1 \le j \le m}$, we define the averaged post-channel non-unitarity as

$$\mathcal{N}_{\text{pc}} = \sum_{j=1}^m p_j \, \mathcal{N}_{\mathcal{U}}(\hat{F}_j, |\psi_{\text{in}}\rangle), \tag{25}$$

with $p_j = \langle \psi_{\text{in}}| \hat{F}_j^\dagger \hat{F}_j |\psi_{\text{in}}\rangle$.

Straightforward calculations allow us to express this quantity directly as

$$\mathcal{N}_{\text{pc}} = -\text{tr}\,\mathbb{1} + \sum_j \frac{\text{tr}\left(\hat{F}_j^\dagger \hat{F}_j \hat{F}_j^\dagger \hat{F}_j\right)}{p_j}. \tag{26}$$

The terms of this sum can be explicitly expressed using the original Kraus operators $\{\hat{E}_j\}$ via the matrix elements of $U$ from Eq. (2):

$$p_j = \sum_{k,l} u_{jk}^* u_{jl} \langle \psi_{\text{in}}| \hat{E}_k^\dagger \hat{E}_l |\psi_{\text{in}}\rangle, \tag{27}$$

$$\text{tr}(\hat{F}_j^\dagger \hat{F}_j \hat{F}_j^\dagger \hat{F}_j) = \sum_{a,b,c,d} u_{ja}^* u_{jb} u_{jc}^* u_{jd} \text{tr}(\hat{E}_a^\dagger \hat{E}_b \hat{E}_c^\dagger \hat{E}_d). \tag{28}$$

Eqs. (26), (27) and (28) are our starting point for a numerical algorithm that adaptively selects the Kraus operator representation, i.e., NUMU. It is interesting to point out the similarity of the expressions (27) and (28) to the results obtained from the effective spin-model mapping in [43], which also leads to expressions depending on the traces $\text{tr}(\hat{F}_j^\dagger \hat{F}_j)$ and $\text{tr}(\hat{F}_j^\dagger \hat{F}_j \hat{F}_j^\dagger \hat{F}_j)$. Moreover, the expression $\text{tr}(\hat{F}_j^\dagger \hat{F}_j \hat{F}_j^\dagger \hat{F}_j)$ is proportional to the exponential of the second Rényi entropy of the Kraus operator $\hat{F}_j$, as e.g., defined in Eq. (115) of [53], a useful quantity to characterize measurement-induced phase transitions.

**Trajectory non-unitarity maximizing unraveling (NUMU):** The idea of NUMU is to adaptively maximize $\mathcal{N}_{\text{pc}}$ as defined in Eq. (26) at each noise application in the QT evolution. In contrast to 2-GEO, it only requires a cost of $\mathcal{O}(\chi^2)$, as it does not involve performing singular value decompositions for each trial unitary $U(\theta, \phi)$. The state dependence appears only in Eq. (27) through the overlaps:

$$o_{k,l} = \langle \psi_{\text{in}}| \hat{E}_k^\dagger \hat{E}_l |\psi_{\text{in}}\rangle = \sum_{i=0}^1 \sum_{\alpha,\beta}^\chi \left(\tilde{A}_{\alpha,i,\beta}^{(k)}\right)^* \tilde{A}_{\alpha,i,\beta}^{(l)}, \tag{29}$$

where $\tilde{A}^{(1,2)}$ denote the normalized tensors $A^{(1,2)}$ defined in Eq. (21). Computing the overlaps $o_{k,l}$ can be performed in $\mathcal{O}(\chi^2)$ operations and needs to be performed only once per noise-channel. For each trial unitary $U$ one can compute the probabilities $p_j$ from Eq. (27) re-using the $o_{k,l}$ values from Eq. (29). Note that also the values for $\text{tr}(\hat{E}_a^\dagger \hat{E}_b \hat{E}_c^\dagger \hat{E}_d)$ in Eq. (28) can be fully pre-calculated at the beginning of the simulation. In practice as optimizer in our simulations we use a standard limited-memory BFGS optimization algorithm.

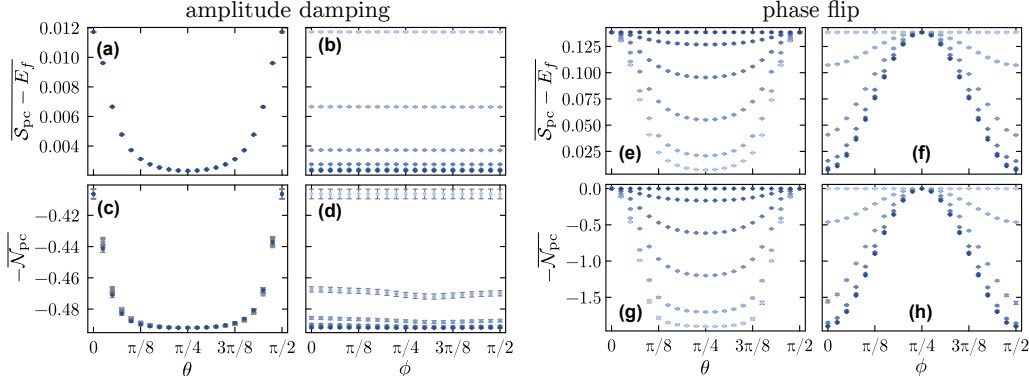

Figure 2: *Comparison between trajectory non-unitarity and post-channel TE* – Kraus operators are applied to the first qubit of random two-qubit states. Plots show the dependence of the excess TE, $\overline{\mathcal{S}_{\mathrm{pc}} - E_f}$ ($E_f$: entanglement of formation) and $-\overline{\mathcal{N}_{\mathrm{pc}}}$ as function of the Kraus operator choice, parametrized by $\theta$ and $\phi$. (a-d): Amplitude damping, (e-h): Phase flip. Different colors correspond to fixed values of $\phi$ (a,c,e,g) and $\theta$ (b, d, f, h), ranging from $\theta, \phi = 0$ (light) to $\pi/4$ (dark) in increments of $\frac{\pi}{20}$. Noise rates: $p_{\mathrm{ad}} = p_{\mathrm{pf}} = 0.1$. Data points averaged over $10^5$ random states. Error bars: standard error of the mean.

**Insight from two qubits systems:** Before going to large-scale simulations, we analyze the connection between $\mathcal{N}_{\mathrm{pc}}$ from Eq. (25) and $\mathcal{S}_{\mathrm{pc}}$ from Eq. (20) using a simple two qubit system. As expected, in Fig. 2 we find a striking anticorrelation between the two quantities. Here, we randomly generate two-qubit states and compute $\mathcal{N}_{\mathrm{pc}}$ as well as $\mathcal{S}_{\mathrm{pc}}$ after applying each of the Kraus operators $\{\hat{F}_j\}_{j=1,2}$ on the first qubit, for various combinations of $\theta$ and $\phi$. Instead of $\mathcal{S}_{\mathrm{pc}}$, we plot the random state averaged excess entropy, $\mathcal{S}_{\mathrm{pc}} - E_f$, subtracting the EoF [54], $E_f$, of the post-channel mixed state. This removes the dependence on the overall entanglement of the randomly sampled state, and it implies that $\mathcal{S}_{\mathrm{pc}} - E_f = 0$ corresponds to the unraveling choice with the lowest possible entanglement. In Fig. 2 we show averaged results for $-\overline{\mathcal{N}_{\mathrm{pc}}}$ and $\overline{\mathcal{S}_{\mathrm{pc}} - E_f}$ using $10^5$ random initial states.

First focusing on amplitude damping, we observe a negligible dependence on the angle $\phi$ and a minimum value attained for $\theta = \pi/4$, which corresponds to $\hat{F}_{\pm}^{\mathrm{ad}} = (\hat{E}_1^{\mathrm{ad}} \pm \hat{E}_2^{\mathrm{ad}})/\sqrt{2}$. This can be understood by deriving the expressions for the two eigenvalues of the reduced density matrix of a single qubit, depending on whether a single Kraus operator $\hat{F}_j^{\mathrm{ad}}$ with $j = 1, 2$ has been applied to one of the qubits, for a general initial state $|\psi_0\rangle$. They read (see Appendix C):

$$\lambda_{\pm}^{j,\mathrm{ad}} = \frac{1}{2} \pm \frac{1}{2}\sqrt{1 - \mathcal{C}_j(\theta, \phi)}, \tag{30}$$

with $C_j(\theta, \phi)$ denoting the concurrence [54] of the post-channel state, itself depending on the input state. To minimize the post-channel entanglement $\mathcal{S}_{\mathrm{pc}}$, we impose that the $+$ eigenvalue be 1 and the $-$ eigenvalue 0, and that this be the case for both Kraus operators $j = 1, 2$. This leads to the condition $\mathcal{C}_1 = \mathcal{C}_2$, satisfied when $\theta = \pi/4$, which suggests this is indeed, on average, the choice with lowest $\mathcal{S}_{\mathrm{pc}}$.

In contrast, for phase flip we find a strong dependence on the phase $\phi$, with the value $\phi = 0$ being favored, while again the optimal values are achieved for $\theta = \pi/4$. This implies, as before, that the Kraus operator choice $\hat{F}_{\pm}^{\mathrm{pf}} = (\hat{E}_1^{\mathrm{pf}} \pm \hat{E}_0^{\mathrm{pf}})/\sqrt{2}$ is on average the best one, consistent with previous findings [42, 43]. For this and similar Pauli channels, we can also derive explicit expressions for $\mathcal{N}_{\mathrm{pc}}$, going beyond two qubit systems as shown below.

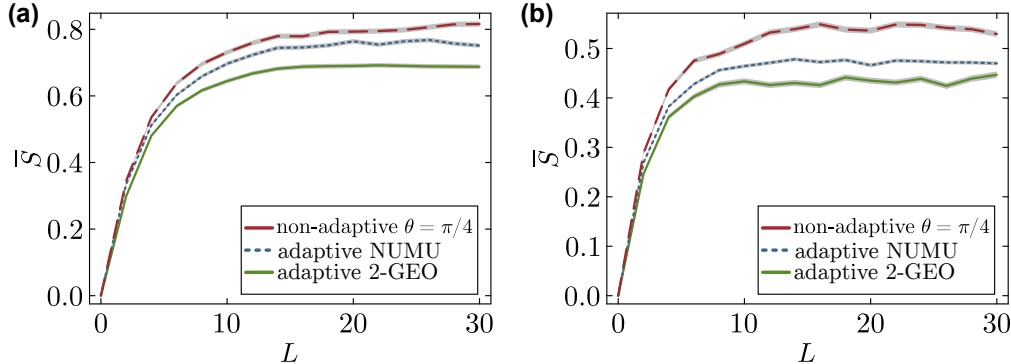

Figure 3: *Small-system comparison between NUMU and 2-GEO* – Evolution of TE, $\overline{S}$ for (a) amplitude damping with $p_{\text{ad}} = 0.3$, and (b) phase flip with $p_{\text{pf}} = 0.1$. NUMU outperforms the non-adaptive unraveling, but is less effective than 2-GEO. Parameters: $n = 30$ qubits, $\chi = 64$, $n_T = 2000$ trajectories, average taken over 5 central bonds, the shaded area represents the standard error of the mean.

Finally, we emphasize that the choices suggested by Fig. 2 are best only when averaged over Haar random states. We will demonstrate that it can be worthwhile to use an adaptive method that takes into account the state at each local noise application instead of fixing the unitary freedom at all sites in the circuit.

**Analytical expression for Pauli channels:**  We can compute the explicit dependence of $\mathcal{N}_{\text{pc}}$ on $\theta$ and $\phi$ from Eq. (26) for the case of Pauli noise channels. The Kraus operators are $\hat{E}_1 = \sqrt{1-p}\mathbb{1}$ and $\hat{E}_2 = \sqrt{p}\hat{\sigma}_\alpha$ for any Pauli operator $\hat{\sigma}_\alpha$ satisfying $\hat{\sigma}_\alpha^\dagger \hat{\sigma}_\alpha = \mathbb{1}$, $\hat{\sigma}_\alpha^2 = \mathbb{1}$ and $\operatorname{tr}\hat{\sigma}_\alpha = 0$. Then, straightforward calculations give (see Appendix D)

$$\mathcal{N}_{\text{pc}}(\theta, \phi; |\psi_{\text{in}}\rangle) = \operatorname{tr}\mathbb{1}\left( \frac{f_1 - f_1^2 + f_2^2 + \langle\psi_{\text{in}}|\hat{\sigma}_\alpha|\psi_{\text{in}}\rangle (f_2 - 2f_1 f_2)}{f_1 - f_1^2 + \langle\psi_{\text{in}}|\hat{\sigma}_\alpha|\psi_{\text{in}}\rangle \left(f_2 - 2f_1 f_2 - \langle\psi_{\text{in}}|\hat{\sigma}_\alpha|\psi_{\text{in}}\rangle f_2^2\right)} - 1 \right), \quad (31)$$

using the two definitions

$$f_1(\theta, p) = (1-p)\cos^2\theta + p\sin^2\theta,$$
$$f_2(\theta, \phi, p) = \sqrt{p - p^2}\sin 2\theta \cos 2\phi.$$

Considering that deep in the random circuit states will be almost randomly distributed according to the Haar measure, we now make the approximation $\langle\psi_{\text{in}}|\hat{\sigma}_\alpha|\psi_{\text{in}}\rangle \approx 0$, as the average expectation value of Pauli operators over the Haar measure is 0. Note that this is a rough approximation replacing the values for $\langle\psi_{\text{in}}|\hat{\sigma}_\alpha|\psi_{\text{in}}\rangle$ by their average. More properly, the average should be taken for the full expression of Eq. (31). Nevertheless, it is useful to develop intuition about why some Kraus operators seem to be so good at lowering entanglement. Note that this then leads to a state-independent expression for $\overline{\mathcal{N}_{\text{pc}}}$ given by

$$\overline{\mathcal{N}_{\text{pc}}}(\theta, \phi) = \frac{(p - p^2)\sin^2(2\theta)\cos^2(2\phi)}{(1-p)\cos^2\theta + p\sin^2\theta - ((1-p)\cos^2\theta + p\sin^2\theta)^2}\operatorname{tr}\mathbb{1}. \quad (32)$$

From this expression, it is clear that the maximum $\mathcal{N}_{\text{pc}}$ is, on average, attained for $\theta \approx \pi/4$ and $\phi \approx 0$. Thus, this expression is in good qualitative agreement with Fig. 2.

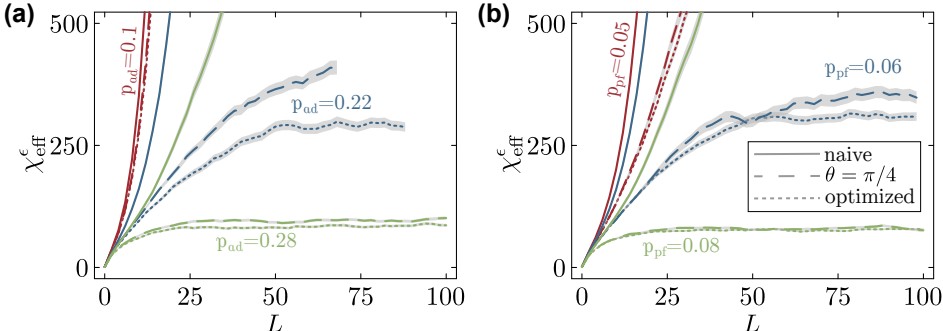

Figure 4: *Effective Schmidt rank for various unravelings* – Evolution of $\chi_{\text{eff}}^{\epsilon}$, defined in Eq. (15), in random circuits with $n = 100$ qubits and $L = 100$ layers for (a) amplitude damping, and (b) phase flip noise. For each panel the error rates, in increasing order, are indicated by the colors: red, blue, and green. For each rate we show different unraveling strategies: naive (solid lines, see text), rotated Kraus operators with $\theta = \pi/4$ and $\phi = 0$ (dashed lines), and using the adaptive NUMU method (dotted lines). The naive unraveling leads to exponential growth of $\chi_{\text{eff}}^{\epsilon}$ in all scenarios, while the other strategies maintain bounded resources, with NUMU achieving a further reduction compared to $\theta = \pi/4$. Parameters: $\chi = 512$, average over $n_T = 250$ trajectories and 21 central bonds, $\epsilon = 10^{-4}$.

**Comparison with 2-GEO:**   To assess the usefulness of NUMU, let us now compare it with the more straightforward, but inefficient, 2-GEO strategy. Fig. 3 shows the dynamics of the TE, $\bar{S}$, averaged over multiple trajectories and Haar-random circuits, in a relatively small, square circuit ($n = 30$ qubits) for the two noise models. Note that the measurement rates are high enough for the entropy to saturate.

As expected, both the adaptive unravelings outperform the best non-adaptive choice of $\theta = \pi/4$ at each noise application and direct optimization outperforms the NUMU approach. Despite the better performance of 2-GEO, we remark that to perform the comparison the circuit must be chosen small enough (in both number of qubits and depth) and the error rate high enough such that direct optimization remains computationally feasible, thus the lower computational cost of NUMU allows for the study of much larger systems as done in the next section.

## 4   Applications to large random quantum circuits

We now analyze the capabilities of NUMU for lowering entanglement for larger scale simulations (Sec. 4.1), in regimes where direct optimization fails, and then provide a comparison with the MPDO approach (Sec. 4.2).

### 4.1   Quantum trajectories

**Comparing unraveling schemes:**   Focusing on the effective Schmidt rank, in Fig. 4(a) we show the evolution of $\chi_{\text{eff}}^{\epsilon}$ in circuits with $n = 100$ qubits undergoing phase flip noise at three different rates $p_{\text{pf}} = 0.05$ (red), $p_{\text{pf}} = 0.06$ (blue), and $p_{\text{pf}} = 0.08$ (green). The three different noise rates lead to two common scenarios: i) an entanglement volume-law regime with exponential growth of the MPS resources as function of the circuit depth $L$, $\chi_{\text{eff}}^{\epsilon} \sim \exp(L)$; or ii) an area-law regime, with saturating entanglement corresponding to constant $\chi_{\text{eff}}^{\epsilon}$. For each rate, the three different line styles (solid, dashed, dotted, from top to bottom) are re-

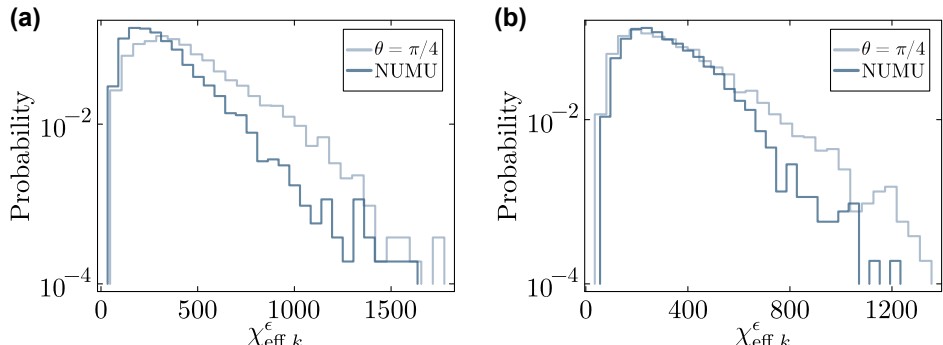

Figure 5: *Effective Schmidt rank distributions* – Normalized histograms of effective Schmidt ranks $\chi^{\epsilon}_{\text{eff},k}$ for $n_T = 250$ trajectories and the 21 central bonds corresponding to data of Fig. 4. The distributions are shown for (a) amplitude damping ($p_{\text{ad}} = 0.22$) after layer $L = 100$ and (b) phase flip ($p_{\text{pf}} = 0.06$) after layer $L = 100$. The comparison is between the optimal non-adaptive unraveling ($\theta = \frac{\pi}{4}$) and NUMU. The NUMU method yields a similar distribution shape, but with a lower mean value. In all cases, the tails decay exponentially. Parameters as in Fig. 4.

sults for 3 different unraveling strategies: i) a "naive" projective unraveling corresponding to Kraus operators $\hat{F}_1 = \sqrt{1-2p}\,\mathbb{1}$, $\hat{F}_2 = \sqrt{2p}\,|0\rangle\langle 0|$, and $\hat{F}_3 = \sqrt{2p}\,|1\rangle\langle 1|$ (solid line); ii) the optimal non-adaptive unraveling with $\theta = \pi/4$, corresponding to the two Kraus operators $\hat{F}_{\pm} = 1/\sqrt{2}\,(\hat{E}^{\text{pf}}_1 \pm \hat{E}^{\text{pf}}_2)$ (dashed line); and iii) adaptive NUMU (dotted line). Note that for the "naive" case, we do not take the Kraus operators $\hat{E}^{\text{pf}}_1$ and $\hat{E}^{\text{pf}}_2$ as defined in Eq. (17), since this would fail to reduce entanglement entirely compared to the noise-less case, given the fact that both $\hat{E}^{\text{pf}}_1$ and $\hat{E}^{\text{pf}}_2$ are (proportional to) unitary matrices.

We also show the same plot for amplitude damping in Fig. 4(b). The respective noise rates are $p_{\text{ad}} = 0.1$ (red), $p_{\text{ad}} = 0.22$ (blue), and $p_{\text{ad}} = 0.28$ (green). Here, the "naive" unraveling now corresponds to using the bare non-unitary Kraus operators defined in Eq. (16).

Comparing the growth of $\chi^{\epsilon}_{\text{eff}}$ across different unravelings shows the performance differences between them. For the naive unraveling, we observe that, for both phase flip and amplitude damping, the effective Schmidt rank grows exponentially with circuit depth $L$, regardless of noise rate. This rapid growth indicates that, even for relatively large error rates $p_{\text{pf}} = 0.08$ and $p_{\text{ad}} = 0.28$, QT simulation become unfeasible beyond a small depth. In contrast, for intermediate values like $p_{\text{pf}} = 0.06$ and $p_{\text{ad}} = 0.22$, both the non-adaptive $\theta = \pi/4$ and NUMU unravelings are able to constrain $\chi^{\epsilon}_{\text{eff}}$ and achieve saturation, making these cases tractable for deeper circuits.

NUMU's performance is underscored by its ability to reduce the effective Schmidt rank compared to non-adaptive strategies. In simulations where $\theta = \pi/4$ and NUMU display a saturating behavior, we observe that NUMU consistently achieves lower values of $\chi^{\epsilon}_{\text{eff}}$. For example, specifically for $p_{\text{ad}} = 0.22$, we observe a reduction by about 25% compared to the non-adaptive strategy. Given the cubic scaling of simulation time with bond dimension in MPS state updates, this reduction translates into a potential doubling of computational speed. In all considered cases and for both noise channels, we always observe a modest lowering of $\chi^{\epsilon}_{\text{eff}}$ by a constant value, leading to an enhanced performance when transitioning from the best non-adaptive strategy to NUMU.

In order for QT+MPS to be able to faithfully describe dynamics, not only the mean value $\chi^{\epsilon}_{\text{eff}}$ is important, but also the distribution of $\chi^{[\nu],\epsilon}_k$ over different trajectories (and bonds). We analyze this distribution in Fig. 5, comparing the cases of a $\theta = \pi/4$ unraveling with NUMU for amplitude damping and phase flip with rates $p_{\text{ad}} = 0.22$ and $p_{\text{pf}} = 0.06$, respectively.



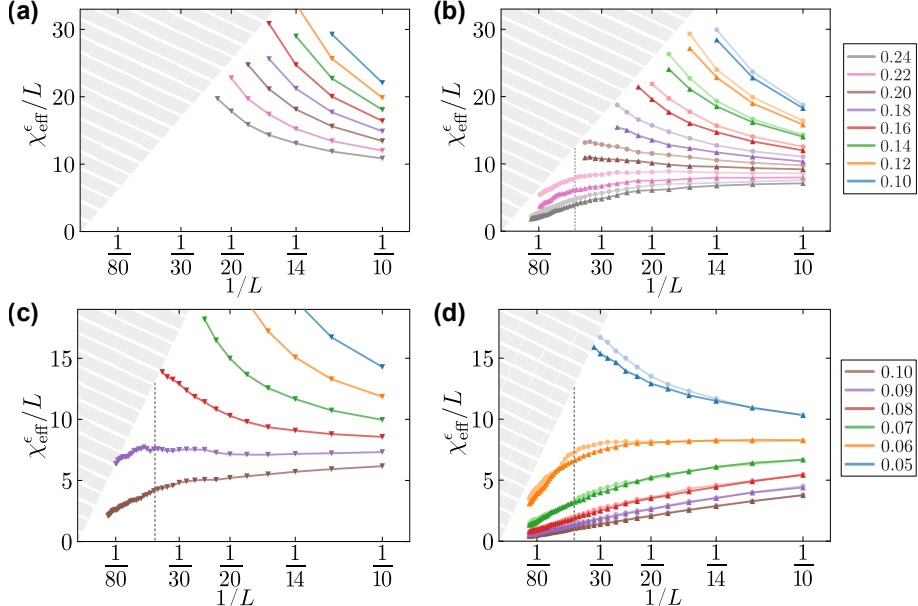

Figure 6: *Determining the crossover between QT-simulable and QT-non-simulable* – Plotted is $\chi^{\epsilon}_{\text{eff}}/L$ as function of $1/L$ for amplitude damping [(a) and (b)] and for phase flip [(c) and(d)]. The left panels (a, c) depict the naive unraveling, which yields higher values of $\chi^{\epsilon}_{\text{eff}}/L$ compared to the lighter lines for $\theta = \pi/4$ and darker lines for NUMU in the right panels (b, d). When dividing by the circuit depth, $\chi^{\epsilon}_{\text{eff}}/L$ grows exponentially in a TE volume-law phase, approaches zero in an area-law phase, and should remain constant in the case of a critical region (with log-growth TE). The upper left triangular area denotes a numerically hard regime defined by $\chi_{\text{eff}} > 500$. Left of vertical dotted lines boundary effects may become important. Results are averaged over multiple trajectories ($n_T = 250$), circuit realizations, and over 21 central bonds. Parameters: $n = 100$, $\chi = 512$, $\epsilon = 10^{-4}$.

The plot (logarithmic scale) indicates a clear exponential decay of the distribution, implying an exponential suppression of highly entangled trajectories, as expected in the QT-simulable regime. We note that the probability distribution function retains a similar shape for both noise channels. Also, between the cases of $\theta = \pi/4$ and NUMU the shape of the distribution does not change, but only shifts to smaller values. This is consistent with the observation that NUMU does not lead to a change of the critical noise rate between a QT-simulable and QT-non-simulable regimes as we will analyze it in the next paragraph. The improvement achieved by NUMU is more pronounced for amplitude damping compared to phase flip noise. Interestingly, when comparing 2-GEO to NUMU (Fig. 3) we found that for amplitude damping the entanglement reduction from NUMU was less efficient. This hints to a large potential for finding better adaptive unraveling schemes, in particular in the case of amplitude damping, where the rates for achieving a QT-simulable regime are also still especially high.

**Determining QT-simulable regimes:**   We are now interested in finding approximate values for the error rates separating the non-simulable and simulable regimes when using QT. We note that in the next section 4.2 we will show that this not necessarily connects to simulability for the whole density matrix evolution.

The behavior of $\chi^{\epsilon}_{\text{eff}}/L$ in Fig. 6 provides a basis for estimating the error rates at which noisy quantum circuit dynamics become simulable with QT. Dividing the effective Schmidt rank by the layer number $L$ allows us to distinguish between different phases: it grows exponentially in

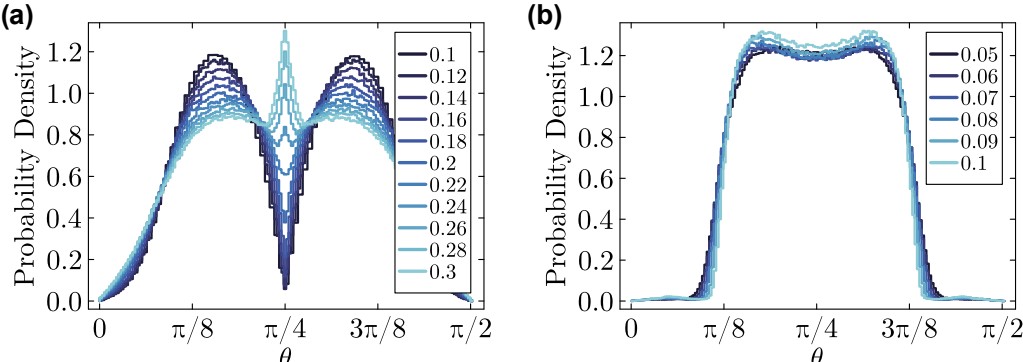

Figure 7: *Rotation angles $\theta$ chosen by the NUMU optimizer* – Histogram of adaptively selected $\theta$ values for maximizing the trajectory non-unitarity for (a) amplitude damping at rates $p_{\mathrm{ad}} = 0.1, \ldots, 0.3$ and (b) phase flip at rates $p_{\mathrm{pf}} = 0.05, \ldots, 0.1$. Interestingly in (a) a peak at $\theta = \pi/4$ appears at the crossover from QT-simulable to QT-non-simulable at $p_{\mathrm{ad}} \approx 0.2$. Parameters as in Fig. 6.

the volume-law phase, approaches zero in the area-law phase, and remains constant in the case of a critical region (with log-growth TE). Fig. 6 shows $\chi_{\mathrm{eff}}^{\epsilon}/L$ as function of $1/L$. The hashed out triangular area in the figure marks a computationally hard region to reach, which we here define as $\chi_{\mathrm{eff}}^{\epsilon} > 500$ (in our simulations all results are converged with a bond dimension of $\chi = 512$). Furthermore, the vertical dotted black line at $(1/L = 1/40)$ indicates a region to the left of which boundary effects can start playing a role. Given the nearest-neighbor connectivity of our random circuit, information in the system can at most spread within a light cone [48]. Here this implies that information from the qubits at the edge of the system reaches the central 21 bonds (used for our averaging) after $L \sim 40$.

Focusing first on amplitude damping in Fig. 6(a/b), we see that for the naive unraveling in panel (a), the critical rate exceeds the highest value considered, suggesting $p_c > 0.24$. In contrast, for the $\theta = \pi/4$ and adaptive NUMU unravelings in panel (b), the curves flatten around $p_c \approx 0.20$. As we have observed it in the examples of Fig. 4, also here in all cases NUMU leads to a lowering of $\chi_{\mathrm{eff}}$ by a constant value. It is interesting to point out that this constant reduction seems to be largest precisely around the critical regime, i.e., for $p_{\mathrm{ad}} = 0.18, 0.20, 0.22$. In other words, we find that NUMU works best in a critical regime. However, it is important to point out that from our pragmatic distinction of QT-simulable vs. QT-non-simulable, we cannot draw conclusions about the existence of a critical regime or a transition. In other words, it may be possible that lines bending upwards for $1/L \to 0$ in Fig. 6 may eventually bend down and go to 0 in the numerically inaccessible regime.

Identical plots for phase flip noise are shown in Fig. 6(c/d), where we observe a similar behavior. Note that the QT-simulable regime stops for the naive unraveling in panel (c) at $p_c \approx 0.09$, while for the $\theta = \pi/4$ and adaptive NUMU in panel (d) the crossover drops to $p_c \approx 0.06$.

Overall, the $\theta = \pi/4$ unraveling performs much better than the naive strategy for both phase flip and amplitude damping noise. From our data we cannot conclude that NUMU leads to a significantly lowering of $p_c$ compared to the non-adaptive rotated Kraus operators. However, we do observe that NUMU leads to the largest reduction in complexity close to $p_c$. As discussed above, the results from the much higher rate of $p_c$ for amplitude damping suggests more room for improvement with a more advanced adaptive method than for phase flip.

Lastly, it turns out to be very interesting to examine the values of the parameters $\theta$ and $\phi$ that are selected by the NUMU optimizer. The choice of $\phi$ is straightforward to understand. For phase flip, the choice is always $\phi \approx 0$, as follows from trying to maximize non-unitarity $\overline{\mathcal{N}_{\mathrm{pc}}}$ in

Eq. (32), specifically the $\cos(2\phi)$ factor. For amplitude damping, it is uniformly distributed, as can be seen in the supplementary material C. Histograms for $\theta$ lead to a much more intriguing behavior as shown in Fig. 7. For phase flip [panel (b)], they cluster around $\theta = \pi/4$, as expected, and independent of the chosen rate $p_{\text{pf}}$. In contrast, for amplitude damping [panel (a)] the behavior changes qualitatively with increasing error rate $p_{\text{ad}}$: at low $p$ values around $\pi/8$ and $3\pi/8$ are preferred, but a peak appears at $\theta = \frac{\pi}{4}$ as $p_{\text{ad}}$ increases. Crucially, this peak occurs in between $p_{\text{ad}} = 0.20$ and $p_{\text{ad}} = 0.22$, which is exactly the regime where we estimated the crossover between QT-simulable and QT-non-simulable regimes. This hints to a clear qualitative change of TE entanglement growth behavior at a critical rate.

## 4.2 Comparison with matrix product density operator simulations

So far we have compared different strategies to unravel the density matrix evolution. Now we will also assess how it compares to the alternative approach where the density matrix itself is decomposed into a Matrix Product Density Operator (MPDO) [15, 20–23]. Here, a decomposition into local tensors similar to the one in the MPS ansatz in Eq. (12) is used

$$
\hat{\rho} = \sum_{\{i_\nu\}} \sum_{\{\alpha_\nu\}} \prod_{\nu=1}^{n} R^{[\nu]i_\nu}_{\alpha_\nu \alpha_{\nu+1}} \lambda^{[\nu],\rho}_{\alpha_\nu} \bigotimes_\nu \hat{e}_{i_\nu}, \tag{33}
$$

where $R^{[n]i_n}_{a_n a_{n+1}}$ are three-dimensional tensors and $\lambda^{[n],\rho}_{a_n}$ are the vectors of normalized Schmidt values fulfilling $\sum_{a_n} (\lambda^{[n],\rho}_{a_n})^2 = 1$. Similarly to MPS, only the $\chi$ largest Schmidt values are retained to obtain an approximate representation of $\hat{\rho}$ [20, 21]. Note that here the approximation is made with respect to the $L^2$ norm of the vectorized density matrix. In order to vectorize the density matrix we have introduced orthonormal basis operators $\hat{e}_{i_n}$, fulfilling $\text{tr}(\hat{e}_i \hat{e}_j) = \delta_{ij}$. We propose to use matrices related to Pauli matrices, and we define them as:

$$
\hat{e}_1 = \frac{1}{\sqrt{2}}\mathbb{1}, \qquad \hat{e}_2 = \frac{1}{\sqrt{2}}\hat{\sigma}^x, \qquad \hat{e}_3 = \frac{1}{\sqrt{2}}\hat{\sigma}^y, \qquad \hat{e}_4 = \frac{1}{\sqrt{2}}\hat{\sigma}^z. \tag{34}
$$

In contrast to the commonly used Choi representation, here the vectorization using the matrices from Eq. (34) has the computational advantage that they lead to the tensors $R^{[n]i_n}_{a_n a_{n+1}}$ being real.

Analogous to pure state MPS decompositions, one can also define an "operator entanglement" (OE)

$$
S^{(n)}_{\text{OP}} = -\sum_{a_n} \left(\lambda^{[n],\rho}_{a_n}\right)^2 \log_2 \left(\lambda^{[n],\rho}_{a_n}\right)^2. \tag{35}
$$

Similar to TE, OE is not a true measure of entanglement. However, it is linked to the efficiency of the decomposition Eq. (33), since for a truncation to bond dimension $\chi$, the OE is limited to values $S_{\text{OP}} \leq \log_2(\chi)$. Using the normalized Schmidt values $\lambda^{[n],\rho}_{a_n}$, instead of relying on OE, we can again define an effective Schmidt rank, $\chi^{\epsilon,\text{MPDO}}_{\text{eff}}$, by simply reusing the definition in Eq. (15). As for pure states, $\chi^{\epsilon,\text{MPDO}}_{\text{eff}}$ offers a rigorous estimate on how large MPDO tensors need to be in order to approximate $\hat{\rho}$ (with respect to the $L^2$ norm) when allowing for a truncation error bounded by $\epsilon$.

Fig. 8 shows dynamics of the effective Schmidt rank for MPDO simulations of our circuits with amplitude damping and phase flip, for the identical parameters as used in the QT+MPS simulations of Fig. 4. As expected, we observe a behavior of exponential rise and fall. Once overcoming the "entanglement barrier", the Schmidt rank decays exponentially for the phase flip channel, i.e., we obtain states with small OE. This can be understood by the fact that for

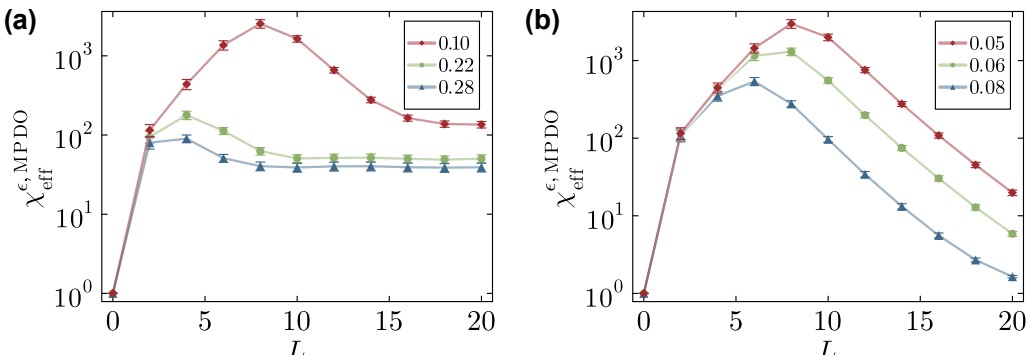

Figure 8: *Effective MPDO Schmidt rank* – Evolution of $\chi_{\text{eff}}^{\epsilon,\text{MPDO}}$ in random circuits with $n = 100$ qubits for the same noise rates and parameters as in Fig. 4. The circuit is subjected to (a) amplitude damping or (b) phase flip noise channels. Results are averaged over 30 random circuit instances, bond dimension is $\chi = 1024$, except for the lowest rate curves where it is $\chi = 2048$.

phase flip the system converges to the maximally mixed state $\propto \mathbb{1}$, which in this case is a left/right eigenstate of the Klaus operators from Eq. (17). In contrast, for amplitude damping $\chi_{\text{eff}}^{\epsilon,\text{MPDO}}$ converges to a computationally cheap value dependent on the rate $p_{\text{ad}}$. In all cases, after applying $L = 20$ layers the effective Schmidt rank is significantly smaller in the MPDO simulations, for example comparing the effective Schmidt rank of MPDO with NUMU for the largest rates, we find for amplitude damping ($p_{\text{ad}} = 0.28$): $\chi_{\text{eff}}^{\epsilon,\text{MPDO}} \approx 40 \lesssim \chi_{\text{eff}}^{\epsilon} = 75$; and for phase flip ($p_{\text{pf}} = 0.08$): $\chi_{\text{eff}}^{\epsilon,\text{MPDO}} \approx 2 \lesssim \chi_{\text{eff}}^{\epsilon} = 75$. This implies that even in the cases where QT+MPS simulations are tractable up to arbitrary depths, the state approximation in terms of a trajectory is inefficient, with trajectories deep in the circuit featuring unnecessary entanglement.

Strikingly, QT+MPS simulations can exhibit a volume law entanglement scaling, leading to exponentially growing effective Schmidt rank in the trajectories, $\chi_{\text{eff}}^{\epsilon} \propto \exp(L)$, while MPDO can overcome the entanglement barrier with a large enough bond dimension cutoff. This implies that the entanglement barrier for $\chi_{\text{eff}}^{\epsilon,\text{MPDO}}$ sets a lower simulability threshold than the exponential scaling of $\chi_{\text{eff}}^{\epsilon}$ in QT+MPS, even when using NUMU. This advantage of MPDO holds for most scenarios [35,36], although counter-examples exist where TE scales more favorably than OE [42]. The computational difficulty in QT+MPS arises from the highly entangled trajectories. For instance, in the case of phase flip noise, $\chi_{\text{eff}}^{\epsilon,\text{MPDO}} \to 1$ as $L$ increases, indicating that the system converges to a trivial separable mixed state with zero EoF. This represents a lower bound for TE in this scenario and suggests untapped potential for lowering entanglement by careful choice of the QT+MPS unraveling. Establishing more precise inequalities between $\chi_{\text{eff}}^{\epsilon}$ in QT+MPS and $\chi_{\text{eff}}^{\epsilon,\text{MPDO}}$ would provide a clearer comparison.

# 5 Conclusion & outlook

In this work, we developed a new adaptive quantum trajectory unraveling method, denoted NUMU, and demonstrated its capabilities for simulating random noisy quantum circuits subjected to single-qubit amplitude damping or phase flip noise. The method works fully numerically and has broad applicability to any type of Markovian open quantum dynamics problem, e.g., also Lindblad master equations. An interesting future direction is to test NUMU in the context of models more structured than Haar random circuits, which constitute a worst-case scenario in terms of large entanglement build-up.

For generic open quantum dynamics, commonly used quantum trajectory algorithms unravel the density matrix into pure trajectory states with a large amount of "entanglement", which makes it numerically hard to use MPS decompositions for such trajectory states. It is therefore desirable to find alternative unraveling strategies with lower averaged trajectory entanglement (TE). In comparison with matrix product density operator (MPDO) simulations, we have found that in our random circuits the TE is excessively large, even for scenarios where the full density matrix is separable.

A strategy to lower TE relies on using the unitary degree of freedom of the Kraus operator representation describing the noise channel. We have introduced a new figure of merit, denoted as "post-channel non-unitarity" $\mathcal{N}_{pc}$ [Eq. (26)], and argued that maximizing $\mathcal{N}_{pc}$ leads to an optimized lowering of the averaged post-channel entanglement entropy, $\mathcal{S}_{pc}$. Using small-scale systems and analytical arguments, we have shown that for random states and amplitude damping or phase flip noise channels, the largest $\mathcal{N}_{pc}$ occurs for Kraus operators that are rotated from their original form [Eqs. (16) and (17)] to $\hat{F}^{ad,pf}_{\pm} = (\hat{E}^{ad,pf}_1 \pm \hat{E}^{ad,pf}_1)/\sqrt{2}$. While this has been known for specific problems [41–43], here we could deduce this from arguments involving $\mathcal{N}_{pc}$. Importantly, in practical simulations, adaptively choosing the Kraus operator unitary degree of freedom, i.e., dependent on the input state, can further increase $\mathcal{N}_{pc}$ and lower $\mathcal{S}_{pc}$. This led us to propose the adaptive non-unitarity maximizing unraveling approach, NUMU.

The key advantage of NUMU is that the on-the-fly numerical maximization of $\mathcal{N}_{pc}$ in an MPS is significantly cheaper than the computation of $\mathcal{S}_{pc}$. For a bond-dimension $\chi$, computing $\mathcal{N}_{pc}$ only costs $\mathcal{O}(\chi^2)$ operations, much lower compared to the $\mathcal{O}(\chi^3)$ cost for calculating $\mathcal{S}_{pc}$. Applying gates in TEBD-style updates also has a computational cost of $\mathcal{O}(\chi^3)$, and therefore for large $\chi$ the NUMU optimization adds a negligible numerical overhead. While NUMU does not achieve the same reduction as a direct minimization of $\mathcal{S}_{pc}$ (Fig. 3) (known in the literature as 2-GEO [42]), we find that it still significantly lowers computational requirements in large-scale calculations, where 2-GEO is too expensive (Fig. 4). For the random quantum circuit simulations considered here, this could lead to a speed-up up to a factor of two. In order to analyze the MPS-simulability, instead of entanglement entropies, we made use of an effective Schmidt rank, which provides a rigorous measure of simulability and does not depend on the chosen entropy measure.

Furthermore, we looked at how the noise rate affects the simulability of the noisy circuits, for different unraveling schemes (Fig. 6). This confirmed that the rate $p_c$, above which the system becomes simulable using a trajectory unraveling depends strongly on the Kraus operator choice. However, using our numerical NUMU approach we could not observe a further shift of $p_c$ compared to the best non-adaptive scheme using the rotated operators $\hat{F}^{ad,pf}_{\pm}$. However, we found that NUMU itself can give interesting new insight into a possible transition at $p_c$ by analyzing the chosen optimization parameters (Fig. 7), and it consistently reduces effective Schmidt ranks by a constant factor.

Importantly, we want to highlight that QT-simulability is not necessarily connected to general simulability. We showed that MPDO simulations can overcome operator entanglement barriers, even in regimes where the TE exhibits volume law scaling. This suggests that there is considerable potential for further optimizing unraveling strategies. Fundamentally, TE is only bounded from below by the entanglement of formation (EoF). The large TE values we find, even when the density matrix is separable, indicate that TE is unnecessarily large. Computing EoF is NP-hard for general mixed states, and thus lowering TE towards the EoF seems intractable. However, using more elaborate schemes, for dynamics starting from a specific initial (typically product) state, may still be able to significantly lower the TE. Investigating this is not only of practical interest, but is also of fundamental interest in the context of our understanding of genuine entanglement in mixed many-body quantum states. So far we have

exclusively optimized Kraus operators for single qubit channels in individual trajectories, and found that this could not shift $p_c$. It would now be very interesting to investigate if more elaborate schemes can achieve such a shift. Here, it would now be important to analyze methods that make use of: i) multiple Kraus channels at the same time; ii) measurement histories ("beyond Markov"); or iii) information of multiple trajectories. The idea of maximizing the averaged post-channel non-unitarity $\mathcal{N}_{\text{pc}}$ provides a new starting point for such investigations.

# Acknowledgments

We thank Tatiana Vovk, Hannes Pichler, and Jérôme Dubail for valuable discussions. Large-scale MPS and MPDO simulations where using the MIMIQ Quantum Circuit Simulation software developed by the start-up QPerfect https://github.com/qperfect-io. FSR thanks everyone at l'Institut de Science et d'Ingénierie Supramoléculaires for the hospitality, especially those at the Centre Européen de Sciences Quantiques. Some of the developed codes make use of the ITensor library [55] and of the Optim package [56].

**Funding information**    Work in Strasbourg is supported by the Interdisciplinary Thematic Institute QMat, as part of the ITI 2021-2028 program of the University of Strasbourg, CNRS and Inserm, and was supported by IdEx Unistra (ANR-10-IDEX-0002), SFRI STRAT'US project (ANR-20-SFRI-0012), and EUR QMAT ANR-17-EURE-0024 under the framework of the French Investments for the Future Program. Work was supported by the CNRS through the EMERGENCE@INC2024 project DINOPARC and by the French National Research Agency under the Investments of the Future Program project ANR-21-ESRE-0032 (aQCess). Computations were carried out using resources of the High Performance Computing Center of the University of Strasbourg, funded by Equip@Meso (as part of the Investments for the Future Program) and CPER Alsacalcul/Big Data.

**Competing interests**    JS is co-founder and shareholder of QPerfect.

# A    Using Chebyshev'sinequality to define an effective Schmidt rank

We start from the probability distribution of squared Schmidt values, $p_\alpha$, and consider the index $\alpha$ as a random variable (bond and trajectory indices are left out for simplicity). Having the mean-index $\mu$ and its standard deviation $\sigma$, we can express the cutoff bond dimension $\chi$ as

$$\chi = \mu + \sigma k(\epsilon), \tag{A.1}$$

where $k(\epsilon) > 1$ is some multiplicative factor, measuring the distance of $\chi$ from the mean index $\mu$. The truncation error $p_{\text{tr}} = \sum_{\alpha > \chi} p_\alpha$ can be expressed as

$$p_{\text{tr}} = \mathcal{P}(\alpha > \chi), \tag{A.2}$$

where $\mathcal{P}(\alpha > \chi)$ denotes the probability of finding the index $\alpha > \chi$. Simple algebra (assuming $\alpha > \mu$) gives then

$$p_{\text{tr}} = \mathcal{P}(\alpha > \mu + \sigma k(\epsilon)) = \mathcal{P}(\alpha - \mu > \sigma k(\epsilon)) \le \frac{1}{k(\epsilon)^2}, \tag{A.3}$$

where in the last line we have used Chebyshev's inequality.

Defining a tolerance $\epsilon$ via $k(\epsilon) = 1/\sqrt{\epsilon}$ we find that the truncation error is bounded $p_{\mathrm{tr}} \leq \epsilon$ when choosing

$$\chi \geq \chi_{\mathrm{eff}} \equiv \mu_k + \frac{\sigma_k}{\sqrt{\epsilon}}. \tag{A.4}$$

The effective Schmidt rank provides an upper bound. As long as a bond dimension cutoff $\chi$ is selected larger or equal than $\chi_{\mathrm{eff}}$, then the truncation error will be $p_{\mathrm{tr}} \leq \epsilon$.

## B  Minimal parametrization for the unitary freedom

For the case of a quantum channel described by two Kraus operators $\{\hat{E}_j\}_{j=1,2}$, the unitary freedom implies that any semi-unitary matrix $U \in \mathcal{M}_{m,2}$ (i.e., such that its first two columns are the same as those of $m \times m$ unitary matrix ($m \geq 2$)) can be used to describe alternative ensembles of Kraus operators $\{\hat{F}_j\}_{j=1,m}$ implementing the same quantum channel as per Eq. (2).

For our single qubit noise channels we focus on a unitary degree of freedom that can be derived from entangling the system qubit with a single ancilla qubit. This restriction is based on the fact that maximal entanglement of a single qubit can be achieved with just a single ancilla, i.e., we use $m = 2$. Convex analysis allows to proof that $m \leq 4$ is sufficient [42]. We can further support our restriction to $m = 2$ by numerical evidence: choosing random representations of unitary matrices of varying sizes does in no case lead to lower minimum attainable entanglement when using $m = 3$ or $m = 4$, as shown in Fig. 9.

For $m = 2$, a general parametrization of a $2 \times 2$ unitary matrix uses 4 angles:

$$U(\theta, \phi, \alpha, \beta) = e^{i\alpha} \begin{pmatrix} e^{i\beta} & 0 \\ 0 & e^{-i\beta} \end{pmatrix} \begin{pmatrix} \cos\theta & \sin\theta \\ -\sin\theta & \cos\theta \end{pmatrix} \begin{pmatrix} e^{i\phi} & 0 \\ 0 & e^{-i\phi} \end{pmatrix}. \tag{B.1}$$

We are interested in the change that the unitary parametrization has on the states:

$$\begin{bmatrix} \hat{F}_1 |\psi_{\mathrm{in}}\rangle \\ \hat{F}_2 |\psi_{\mathrm{in}}\rangle \end{bmatrix} = \begin{pmatrix} e^{i(\alpha+\beta)} & 0 \\ 0 & e^{i(\alpha-\beta)} \end{pmatrix} \begin{pmatrix} \cos\theta & \sin\theta \\ -\sin\theta & \cos\theta \end{pmatrix} \begin{pmatrix} e^{i\phi} & 0 \\ 0 & e^{-i\phi} \end{pmatrix} \begin{bmatrix} \hat{E}_1 |\psi_{\mathrm{in}}\rangle \\ \hat{E}_2 |\psi_{\mathrm{in}}\rangle \end{bmatrix} \tag{B.2}$$

$$\equiv \begin{bmatrix} e^{i(\alpha+\beta)} |\phi_1\rangle \\ e^{i(\alpha-\beta)} |\phi_2\rangle \end{bmatrix}, \tag{B.3}$$

where we defined non-normalized states $|\phi_{1,2}\rangle$ that only depend on $\theta$ and $\phi$. The post-channel averaged entropy $\mathcal{S}_{\mathrm{pc}}$, is a direct function of the renormalized states:

$$\mathcal{S}_{\mathrm{pc}} = \sum_j p_j \, S\left(|\phi_j\rangle / \sqrt{p_j}\right), \tag{B.4}$$

with $p_j \equiv \langle\phi_j|\phi_j\rangle$. Since the entropy of the states, $S(|\phi_{1,2}\rangle)$, and $p_j$ does not depend on $\alpha$ and $\beta$, we can neglect those phases and use

$$U(\theta, \phi) = \begin{pmatrix} \cos\theta & \sin\theta \\ -\sin\theta & \cos\theta \end{pmatrix} \begin{pmatrix} e^{i\phi} & 0 \\ 0 & e^{-i\phi} \end{pmatrix}. \tag{B.5}$$

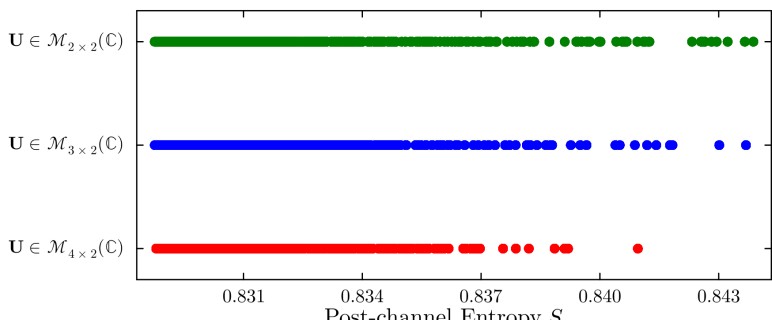

Figure 9: Post-channel averaged entropy $\mathcal{S}_{\mathrm{pc}}$ [Eq. (20)], for amplitude damping Kraus operators [Eq. (16)] that are rotated with random $m \times 2$ semi-unitary matrices $\mathbf{U}$, and applied to the first qubit of a random state of two qubits. We show results for 1000 random configurations. The smallest values of $\mathcal{S}_{\mathrm{pc}}$ are already achieved for $m = 2$.

## C  Optimal parametrization for amplitude damping

Consider an arbitrary state of 2 qubits:

$$|\psi_0\rangle = a\,|00\rangle + b\,|01\rangle + c\,|10\rangle + d\,|11\rangle .$$

We have a quantum channel described by two Kraus operators $\hat{E}_1$ and $\hat{E}_2$ parametrized by two angles $\theta, \phi$ as in B to obtain alternative equivalent operator sum representations:

$$\begin{pmatrix} \hat{F}_1 \\ \hat{F}_2 \end{pmatrix} = \begin{pmatrix} \alpha_1(\theta,\phi) & \beta_1(\theta,\phi) \\ \alpha_2(\theta,\phi) & \beta_2(\theta,\phi) \end{pmatrix} \begin{pmatrix} \hat{E}_1 \\ \hat{E}_2 \end{pmatrix} .$$

For the amplitude damping channel we have the operator sum representation:

$$\hat{E}_1 = \begin{pmatrix} 1 & 0 \\ 0 & \sqrt{1-p} \end{pmatrix} , \qquad \hat{E}_2 = \begin{pmatrix} 0 & \sqrt{p} \\ 0 & 0 \end{pmatrix} .$$

The two generic operators that can represent this channel are thus:

$$\hat{F}_j = \begin{pmatrix} \alpha_j & \sqrt{p}\beta_j \\ 0 & \sqrt{1-p}\alpha_j \end{pmatrix} = \alpha_j\,|0\rangle\langle 0| + \sqrt{p}\beta_j\,|0\rangle\langle 1| + \sqrt{1-p}\,\alpha_j\,|1\rangle\langle 1| ,$$

where $j = 1, 2$ indexes over the two operators, obtained by using the first or second row respectively of the transformation unitary matrix.

We are interested in the entanglement properties of the state (after normalization) obtained by applying the $\hat{F}_j$ operators to the first qubit (for example) of the initial state $|\psi_0\rangle$:

$$\left| \widetilde{\psi}_j \right\rangle \equiv \left( \mathbb{1} \otimes \hat{F}_j \right) |\psi_0\rangle \tag{C.1}$$

$$= \left( a\alpha_j + \sqrt{p}b\beta_j \right) |00\rangle + \sqrt{1-p}\,b\alpha_j\,|01\rangle + \left( c\alpha_j + \sqrt{p}d\beta_j \right) |10\rangle + d\sqrt{1-p}\,\alpha_j\,|11\rangle .$$

The norm of this non-normalized states is given by:

$$\begin{aligned} \left\langle \widetilde{\psi}_j \middle| \widetilde{\psi}_j \right\rangle &= \langle \psi_0 | \hat{F}_j^{\dagger} \hat{F}_j | \psi_0 \rangle \\ &= |a|^2 |\alpha_j|^2 + p\,|b|^2 |\beta_j|^2 + (1-p)|b|^2 |\alpha_j|^2 + |c|^2 |\alpha_j|^2 \\ &\quad + (1-p)|d|^2 |\alpha_j|^2 + p\,|d|^2 |\beta_j|^2 + 2\sqrt{p}\,\mathrm{Re}\left\{ (a^*b + c^*d)\,\alpha_j^*\beta_j \right\} \\ &= |\alpha_j|^2 \left( |a|^2 + |c|^2 \right) + \left( (1-p)|\alpha_j|^2 + p\,|\beta_j|^2 \right) \left( |b|^2 + |d|^2 \right) + 2\sqrt{p}\,\mathrm{Re}\left\{ (a^*b + c^*d)\,\alpha_j^*\beta_j \right\} . \end{aligned}$$

The pure state after the stochastic update with $\hat{F}_j$ is given by the density matrix:

$$\hat{\psi}_j = \left|\psi_j\right\rangle\!\left\langle\psi_j\right| = \frac{|\widetilde{\psi}_j\rangle\langle\widetilde{\psi}_j|}{\langle\widetilde{\psi}_j|\widetilde{\psi}_j\rangle} = \frac{\hat{F}_j\left|\psi_0\right\rangle\left\langle\psi_0\right|\hat{F}_j^\dagger}{\langle\psi_0|\hat{F}_j^\dagger\hat{F}_j|\psi_0\rangle}\,.$$

To understand the entanglement properties of the post-channel pure states we must look at the eigenvalues of the reduced density matrix for the first qubit:

$$\hat{\rho}_j = \mathrm{tr}_2\,\hat{\psi}_j = \sum_{q=0}^{1}(\langle q|\otimes\mathbb{1})\,\hat{\psi}_j\,(|q\rangle\otimes\mathbb{1})$$

$$= \langle 0|_2\,\hat{\psi}_j\,|0\rangle_2 + \langle 1|_2\,\hat{\psi}_j\,|1\rangle_2\,.$$

It is straightforward to compute those two matrices one by one:

$$\langle\widetilde{\psi}_j|\widetilde{\psi}_j\rangle\langle 0|_2\,\hat{\psi}_j\,|0\rangle_2 = \left((a\alpha_j + \sqrt{p}\,b\beta_j)|0\rangle + \sqrt{1-p}\,b\alpha_j|1\rangle\right)\left((a^*\alpha_j^* + \sqrt{p}\,b^*\beta_j^*)\langle 0| + \sqrt{1-p}\,b^*\alpha_j^*\langle 1|\right)$$

$$= \begin{pmatrix} |a|^2\left|\alpha_j\right|^2 + p\,|b|^2\left|\beta_j\right|^2 + 2\sqrt{p}\,\mathrm{Re}\{ab^*\alpha_j\beta_j^*\} & \sqrt{1-p}\,b^*\alpha_j^*\left(a\alpha_j + \sqrt{p}\,b\beta_j\right) \\ \text{c.c. (01)} & (1-p)\,|b|^2\left|\alpha_j\right|^2 \end{pmatrix},$$

$$\langle\widetilde{\psi}_j|\widetilde{\psi}_j\rangle\langle 1|_2\,\hat{\psi}_j\,|1\rangle_2 = \left((c\alpha_j + \sqrt{p}\,d\beta_j)|0\rangle + \sqrt{1-p}\,d\alpha_j|1\rangle\right)\left((c^*\alpha_j^* + \sqrt{p}\,d^*\beta_j^*)\langle 0| + \sqrt{1-p}\,d^*\alpha_j^*\langle 1|\right)$$

$$= \begin{pmatrix} |c|^2\left|\alpha_j\right|^2 + p\,|d|^2\left|\beta_j\right|^2 + 2\sqrt{p}\,\mathrm{Re}\{cd^*\alpha_j\beta_j^*\} & \sqrt{1-p}\,d^*\alpha_j^*\left(c\alpha_j + \sqrt{p}\,d\beta_j\right) \\ \text{c.c. (01)} & (1-p)\,|d|^2\left|\alpha_j\right|^2 \end{pmatrix},$$

where c.c. (01) denotes the complex conjugate of the 01 entry of the matrix. Let us adopt the following notations to simplify further calculations:

$$x \equiv |a|^2 + |c|^2\,, \qquad y \equiv |b|^2 + |d|^2\,, \qquad z \equiv a^*b + c^*d\,.$$

Using this convention we can rewrite the previous norm as:

$$\left\langle\widetilde{\psi}_j\middle|\widetilde{\psi}_j\right\rangle = x\left|\alpha_j\right|^2 + py\left|\beta_j\right|^2 + (1-p)y\left|\alpha_j\right|^2 + 2\sqrt{p}\,\mathrm{Re}\left\{z\alpha_j^*\beta_j\right\},$$

and the resulting reduced density matrix as

$$\hat{\rho}_j = \frac{1}{\left\langle\widetilde{\psi}_j\middle|\widetilde{\psi}_j\right\rangle}\begin{pmatrix} \left\langle\widetilde{\psi}_j\middle|\widetilde{\psi}_j\right\rangle - (1-p)y\left|\alpha_j\right|^2 & \sqrt{1-p}\,\alpha_j^*\left(z^*\alpha_j + \sqrt{p}\,y\beta_j\right) \\ \text{c.c. (01)} & (1-p)y\left|\alpha_j\right|^2 \end{pmatrix}.$$

We remark that the density matrix has the form $\hat{\rho}_j = \frac{1}{\mathrm{tr}\,\tilde{\rho}_j}\,\tilde{\rho}_j$, and it is enough to find the eigenvalues of $\tilde{\rho}_j$ and scale them appropriately. The eigenvalues of a $2\times 2$ density matrix can be expressed using the trace and determinant. Thus, we still need to compute the determinant:

$$\det\tilde{\rho}_j = (1-p)y\left|\alpha_j\right|^2\left(x\left|\alpha_j\right|^2 + py\left|\beta_j\right|^2 + 2\sqrt{p}\,\mathrm{Re}\left\{z\alpha_j^*\beta_j\right\}\right)$$

$$- (1-p)\left|\alpha_j\right|^2\left(z^*\alpha_j + \sqrt{p}\,y\beta_j\right)\left(z\alpha_j^* + \sqrt{p}\,y\beta_j^*\right)$$

$$= (1-p)\left|\alpha_j\right|^4\left(xy - |z|^2\right)$$

$$= (1-p)\left|\alpha_j\right|^4\left(|a|^2\,|d|^2 + |c|^2\,|b|^2 - 2\,\mathrm{Re}\{ab^*c^*d\}\right)$$

$$= (1-p)\left|\alpha_j\right|^4\frac{\mathcal{C}_0^2}{4}\,,$$

where $\mathcal{C}_0 \equiv 2\,|ad - bc|$ is the concurrence of the initial state $|\psi_0\rangle$.

The concurrence is a measure of entanglement in its own right. It takes the value 0 for a product state and the value 1 for a maximally entangled state of two qubits. Moreover, for pure states the von Neumann entanglement entropy can be expressed directly as a monotone and convex function of the concurrence, precisely by using its role in the expression for the eigenvalues of the reduced density matrix. [54]

The eigenvalues of any $2 \times 2$ matrix can be expressed in terms of the determinant and the trace as

$$\lambda_\pm^A = \frac{1}{2}\left(\mathrm{tr}(A) \pm \sqrt{(\mathrm{tr}(A))^2 - 4\det(A)}\right).$$

In the case of the matrices we are studying, the trace is positive, and therefore

$$\lambda_\pm^{\tilde{\rho}_j} = \frac{\mathrm{tr}(\tilde{\rho}_j)}{2}\left(1 \pm \sqrt{1 - \frac{4\det(\tilde{\rho}_j)}{\mathrm{tr}^2(\tilde{\rho}_j)}}\right).$$

Thus, scaling by $\mathrm{tr}\,\tilde{\rho}_j$, the eigenvalues for our reduced density matrices $\hat{\rho}_j$ are then given by

$$\begin{aligned}
\lambda_\pm^j &= \frac{1}{2}\left(1 \pm \sqrt{1 - \frac{4\det(\tilde{\rho}_j)}{\langle\psi_0|\hat{F}_j^\dagger\hat{F}_j|\psi_0\rangle^2}}\right) \\
&= \frac{1}{2}\left(1 \pm \sqrt{1 - \frac{(1-p)|\alpha_j|^4 \mathcal{C}_0^2}{\left(|\alpha_j|^2 + py\left(|\beta_j|^2 - |\alpha_j|^2\right) + 2\sqrt{p}\,\mathrm{Re}\left\{z\alpha_j^*\beta_j\right\}\right)^2}}\right),
\end{aligned}$$

where we used the fact that $x + y = 1$, by the normalization of the initial pure state $|\psi_0\rangle$.

We remark that the second term inside the square root is the concurrence $\mathcal{C}_j$ of the updated state $|\psi_j\rangle \equiv \hat{F}_j|\psi_0\rangle / \sqrt{\langle\psi_0|\hat{F}_j^\dagger\hat{F}_j|\psi_0\rangle}$. This follows directly from the definition of the concurrence and the expression for the states $|\tilde{\psi}_j\rangle$ from Eq. (C.1).

$$\mathcal{C}_j^2 = \frac{(1-p)|\alpha_j|^4 \mathcal{C}_0^2}{\langle\psi_0|\hat{F}_j^\dagger\hat{F}_j|\psi_0\rangle^2}. \tag{C.2}$$

By replacing the explicit dependence on the parameters $\theta$ and $\phi$ from the coefficients $\alpha$ and $\beta$ we find explicitly the eigenvalues of the reduced density matrix, for the cases where $\hat{F}_1$ or $\hat{F}_2$ are applied respectively:

$$\lambda_\pm^1 = \frac{1}{2} \pm \frac{1}{2}\sqrt{1 - \frac{(1-p)\cos^4\theta\,\mathcal{C}_0^2}{\left(\cos^2\theta - py\cos(2\theta) + \sqrt{p}\sin(2\theta)\mathrm{Re}\left\{ze^{-2i\phi}\right\}\right)^2}},$$

$$\lambda_\pm^2 = \frac{1}{2} \pm \frac{1}{2}\sqrt{1 - \frac{(1-p)\sin^4\theta\,\mathcal{C}_0^2}{\left(\sin^2\theta + py\cos(2\theta) - \sqrt{p}\sin(2\theta)\mathrm{Re}\left\{ze^{-2i\phi}\right\}\right)^2}}.$$

The post channel operator averaged entropy will be

$$\overline{S}_{\mathrm{pc}} = -\sum_{j=1}^{2} \langle\psi_0|\hat{F}_j^\dagger\hat{F}_j|\psi_0\rangle\left(\lambda_-^j \log_2 \lambda_-^j + \lambda_+^j \log_2 \lambda_+^j\right).$$

It is clear that for each state $|\psi_j\rangle$ considered separately, the eigenvalue distribution with minimum entropy is $\lambda_+^j = 1$ and $\lambda_-^j = 0$ and the one for maximum entropy $\lambda_+^j = \lambda_+^j = 1/2$.

One simple way to address the question of what values of $\theta$ and $\phi$ minimize $S_{\text{pc}}$ is to use the fact that for two qubit states one optimal decomposition (i.e., for which the average entanglement is the entanglement of formation) of a density matrix into pure states such that the concurrence or entanglement of the two states is equal [57]. Since the two states $\left|\psi_j\right\rangle$ are decomposition of the mixed state obtained by applying the amplitude damping channel to the first qubit of the state $|\psi_0\rangle$, we can simply impose $C_1 = C_2$ which leads to

$$\tan^2\theta = \frac{\sin^2\theta + \cos(2\theta)yp - \sqrt{p}\sin(2\theta)\operatorname{Re}\{e^{i\phi}z\}}{\cos^2\theta - \cos(2\theta)yp + \sqrt{p}\sin(2\theta)\operatorname{Re}\{e^{i\phi}z\}}, \tag{C.3}$$

under the assumptions that $\theta \neq \pi/2$ and $C_0 > 0$ (i.e., $|\psi\rangle$ is not a product state).

It is easy to see that $\theta = \pi/4$ fulfills this condition as long as the state is such that $z \equiv a^*b + c^*d \approx 0$. When averaging over random states this $z$ will approach zero, but otherwise it is easy to see that in general the best parametrization will fulfill the property

$$\cos(2\theta)py = \sin(2\theta)\sqrt{p}\operatorname{Re}\{e^{i\phi}z\}. \tag{C.4}$$

## D  Kraus operators proportional to Pauli matrices

Consider the case of a quantum channel described by the Kraus operators:

$$\hat{E}_1 = \sqrt{1-p}\,\mathbb{1}, \tag{D.1}$$

$$\hat{E}_2 = \sqrt{p}\,\hat{\sigma}, \tag{D.2}$$

with $\hat{\sigma}$ one of the Pauli matrices. This corresponds to the quantum channels Phase Flip, Bit Flip or Bit-Phase Flip.

In this case the operator $\hat{Q}$ from Eq. (22) is trivially state independent and the non-unitarity is zero $\mathcal{N}_{\text{pc}} = 0$. Nonetheless, for transformed Kraus operators $\{\hat{F}\}_{j=1,2}$ from Eq. (2) the non-unitarity can be non-zero. Let us investigate the value for which the maximum is obtained. Eq. (26) shows that it is enough to compute the terms $\hat{F}^\dagger\hat{F}\hat{F}^\dagger\hat{F}$ to obtain $\mathcal{N}_{\text{pc}}$.

We start by writing explicitly those terms for the 2 transformed Kraus operators:

$$\hat{F}_1 = \sqrt{1-p}\cos\theta\, e^{i\phi}\mathbb{1} + \sqrt{p}\sin\theta\, e^{-i\phi}\hat{\sigma}, \qquad \hat{F}_2 = -\sqrt{1-p}\sin\theta\, e^{i\phi}\mathbb{1} + \sqrt{p}\cos\theta\, e^{-i\phi}\hat{\sigma},$$

$$\hat{F}_1^\dagger\hat{F}_1 = \left((1-p)\cos^2\theta + p\sin^2\theta\right)\mathbb{1} \qquad \hat{F}_2^\dagger\hat{F}_2 = \left(p\cos^2\theta + (1-p)\sin^2\theta\right)\mathbb{1},$$
$$\qquad + \sqrt{p-p^2}\sin2\theta\cos2\phi\,\hat{\sigma}, \qquad\qquad - \sqrt{p-p^2}\sin2\theta\cos2\phi\,\hat{\sigma}$$

$$p_1 = \langle\psi_{\text{in}}|\hat{F}_1^\dagger\hat{F}_1|\psi_{\text{in}}\rangle \qquad\qquad p_2 = \langle\psi_{\text{in}}|\hat{F}_2^\dagger\hat{F}_2|\psi_{\text{in}}\rangle$$
$$= f_1 + sf_2, \qquad\qquad\qquad = 1 - f_1 - sf_2,$$

$$\hat{F}_1^\dagger\hat{F}_1\hat{F}_1^\dagger\hat{F}_1 = f_1^2\mathbb{1} + 2f_1f_2\hat{\sigma} + f_2^2\hat{\sigma}^2, \qquad \hat{F}_2^\dagger\hat{F}_2\hat{F}_2^\dagger\hat{F}_2 = (1-f_1)^2\mathbb{1} - 2f_1f_2\hat{\sigma} + f_2^2\hat{\sigma}^2,$$

where we defined:

$$f_1(\theta;p) = (1-p)\cos^2\theta + p\sin^2\theta,$$
$$f_2(\theta,\phi;p) = \sqrt{p-p^2}\sin2\theta\cos2\phi,$$
$$s = \langle\psi_{\text{in}}|\hat{\sigma}|\psi_{\text{in}}\rangle.$$

Now using Eq. (26) and the fact that for a Pauli matrix we have $\operatorname{tr}\hat{\sigma} = 0$ and $\operatorname{tr}\hat{\sigma}^2 = \operatorname{tr}\mathbb{1}$ we can find the dependence of the averaged non-unitarity on the unitary parameters:

$$
\begin{aligned}
\mathcal{N}_{\text{pc}} &= -\operatorname{tr}\mathbb{1} + \sum_j \frac{\operatorname{tr}\left(F_j^\dagger F_j F_j^\dagger F_j\right)}{p_j} \\
&= -\operatorname{tr}\mathbb{1} + \operatorname{tr}\mathbb{1}\frac{f_1^2 + f_2^2}{f_1 + s f_2} + \operatorname{tr}\mathbb{1}\frac{(1-f_1)^2 + f_2^2}{1 - f_1 - s f_2} \\
&= -\operatorname{tr}\mathbb{1} + \operatorname{tr}\mathbb{1}\frac{f_1 - f_1^2 + s f_2 - 2 s f_1 f_2 + f_2^2}{f_1 - f_1^2 + s f_2 - 2 s f_1 f_2 - s^2 f_2^2}\,.
\end{aligned}
$$

It is straightforward to see that in this formula, the unchanged Kraus operators corresponding to $\theta = 0$ and $\phi = 0$ lead to $f_2 = 0$ and therefore to $\overline{\mathcal{N}_{\text{pc}}} = 0$, so to worst-case situation, as we already established.

Please note that as defined above $\overline{\mathcal{N}_{\text{pc}}}$ is an extensive quantity. That is, $\operatorname{tr}\mathbb{1} = 2^n$, with $n$ the number of qubits. Even if the Kraus operators we consider act non-trivially only on single qubit subspace, we need to consider their Kronecker product with appropriately sized identity for obtaining the correct numerical values. Nonetheless, when considering matrix product states as opposed to state vector representations we can safely ignore those factors when considering the maximization problem.

Arguing that the average of $\langle \psi_{\text{in}} | \hat{\sigma} | \psi_{\text{in}} \rangle$ over Haar random states is 0, we can neglect the state dependent terms in the expression and maximize the quantity:

$$
\overline{\mathcal{N}_{\text{pc}}}(\theta, \phi) = \operatorname{tr}\mathbb{1}\left(\frac{f_1 - f_1^2 + f_2^2}{f_1 - f_1^2} - 1\right) = \frac{\operatorname{tr}\mathbb{1} f_2^2}{f_1 - f_1^2}\,.
$$

It is straightforward to see that $\theta = \frac{\pi}{4}$ is an extreme value by checking its derivative:

$$
\begin{aligned}
\frac{\partial \overline{\mathcal{N}_{\text{pc}}}}{\partial \theta} &= \frac{f_2}{\left(f_1 - f_1^2\right)^2}\left(2\left(f_1 - f_2^2\right)\frac{\partial f_2}{\partial \theta} - f_2(1 - 2f_1)\frac{\partial f_1}{\partial \theta}\right) \\
&= \frac{f_2}{\left(f_1 - f_1^2\right)^2}\left(4\left(f_1 - f_2^2\right)\sqrt{p - p^2}\cos 2\theta \cos 2\phi - f_2(1 - 2f_1)(2p - 1)\sin 2\theta\right).
\end{aligned}
$$

From $f_1\left(\theta = \frac{\pi}{4}\right) = \frac{1}{2}$ and $f_2\left(\theta = \frac{\pi}{4}\right) = \sqrt{p - p^2}\cos 2\phi$ it follows directly that the derivative vanishes:

$$
\frac{\partial \overline{\mathcal{N}_{\text{pc}}}}{\partial \theta}\left(\theta = \frac{\pi}{4}\right) = 0\,.
$$

# E  Numerical convergence in the bond dimension

In this appendix, we support the main finding of our work by showing that the results are converged in the bond dimension $\chi$. Convergence behavior may depend on the quantity of interest. Specifically, we focus on the distribution of the effective Schmidt rank $\chi_{\text{eff}}^\epsilon$ across multiple noise rates for both QT+MPS and MPDO simulations.

For the QT, where the average is over multiple trajectories and circuit realizations, we choose as a criterion for convergence whether, after doubling the bond dimension, e.g., from $\chi = 256$ to $\chi = 512$, the standard error of the two curves overlap. The convergence plots are shown in Fig. 10 and in Fig. 11, for two (relatively low) rates $p_{\text{ad}} = 0.1, 0.22$, and

Table 1: Table of the maximum depth that results are converged for different noise channels, unravelings and error rates. Note that for higher rates $p$ all results are converged.

|  | unraveling | phase flip | | amplitude damping | |
|---|---|---|---|---|---|
|  |  | $p = 0.05$ | $p = 0.06$ | $p = 0.1$ | $p = 0.22$ |
| | naive | 20 | 25 | 15 | 25 |
| layer $L$ | $\theta = \pi/4$ | 30 | 100 | 20 | 70 |
| | NUMU | 30 | 100 | 20 | 90 |

$p_{\mathrm{pf}} = 0.05, 0.06$, respectively. They lead to the depth $L$ cutoffs in tables 1. We respect this in all the figure in the main text.

For our MPDO simulations, where the average is only over different circuit realizations, we show convergence in bond dimension in Fig. 12 for both amplitude damping and phase flip. We also compare with the operator "entanglement entropies" and see that they converge faster than the effective Schmidt rank $\chi_{\mathrm{eff}}^\epsilon$, defined in Eq. (15). Note that $\chi_{\mathrm{eff}}^\epsilon$ is thus a more rigorous condition than entanglement entropies for MPS simulability of given state. Note that for MPDO simulations we simulate up to $\chi = 2048$ to find converging behavior in the Schmidt rank, while corresponding von Neumann OE is already clearly captured for $\chi = 512$.

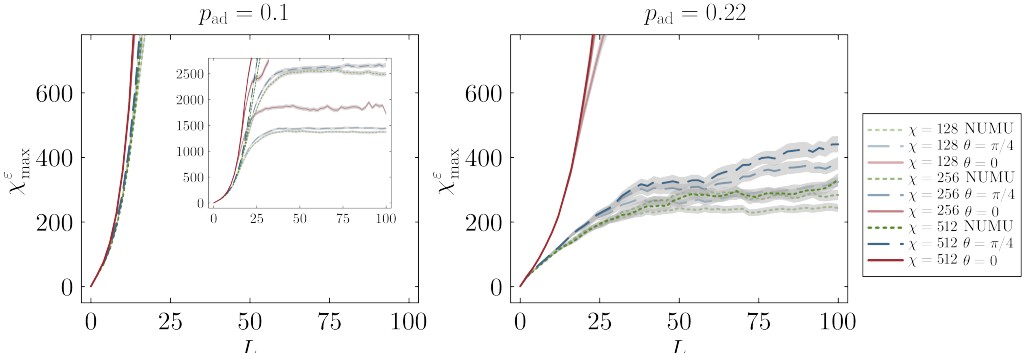

Figure 10: Convergence plots for the amplitude damping channel. The evolution of the effective Schmidt rank for progressively increasing bond dimension, from light to dark $\chi = 128, 256, 512$. Different colors correspond to the different unraveling schemes: "naive" ($\theta = 0$, red), best non-adaptive ($\theta = \pi/4$, blue), and NUMU (green). The gray shaded area correspond to the standard error of the mean. Parameters as in Fig. 4.

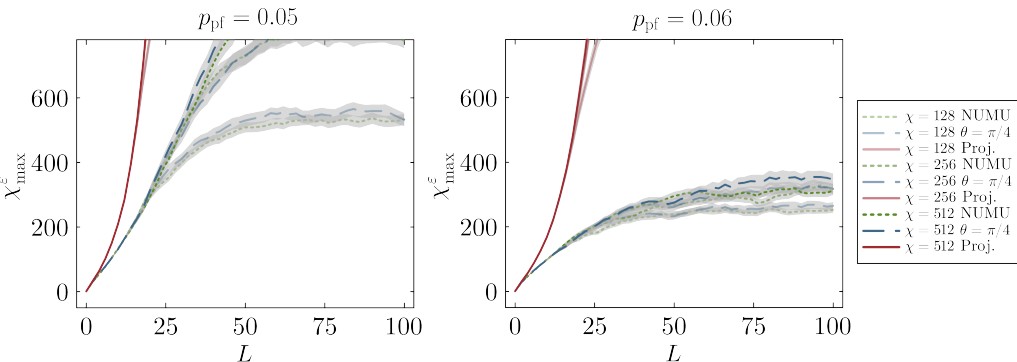

Figure 11: Convergence plots for the phase flip channel. The evolution of the effective Schmidt rank for progressively increasing bond dimension, from light to dark $\chi = 128, 256, 512$. Different colors correspond to the different unraveling schemes: "naive" ($\theta = 0$, red), best non-adaptive ($\theta = \pi/4$, blue), and NUMU (green). The gray shaded area correspond to the standard error of the mean. Parameters as in Fig. 4.

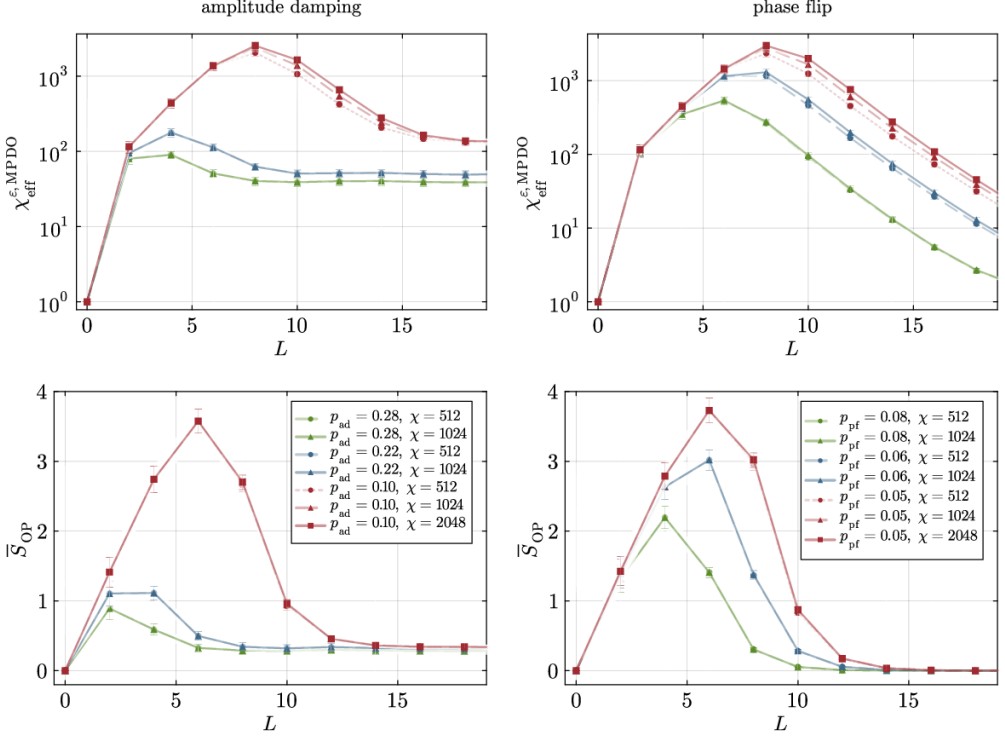

Figure 12: Convergence plots for the MPDO simulations of random circuits for the error rates used in Figs. 1 and 8 in the main text. We show them for both the effective Schmidt ranks, $\chi_{\mathrm{eff}}^{\epsilon,\mathrm{MPDO}}$ and the averaged von Neumann operator entropy $\overline{S}_{\mathrm{OP}}$. The averaging is performed over 10 random quantum circuits. Lines with different colors correspond to different noise rates. Different line styles correspond to bond dimension cutoffs $\chi = 512, 1024, 2048$. Note that the convergence is much faster (smaller $\chi$ needed) for $\overline{S}_{\mathrm{OP}}$ than for $\chi_{\mathrm{eff}}^{\epsilon,\mathrm{MPDO}}$. Parameters as in Fig. 8.

## F  A pseudocode implementation of NUMU

```
1  psi ← |0>
2  E ← kraus_operators_for_chosen_noise()
3  for a, b, c, d in 1:dim(E)  # precompute trace in Eq.(28)
4      trEEEE[a, b, c, d] ← tr(E[a]† * E[b] * E[c]† * E[d])
5  endfor
6
7  for layer in random_circuit
8      for haar_gate in layer  # apply unitary gates
9          psi ← haar_gate * psi
10     endfor
11     for qubit in 1:n  # apply noise to every qubit
12         overlaps ← precompute_overlaps(psi, E)
13         cost_function(theta, phi) ← \
14             non_unitarity_cost(theta, phi, overlaps, trEEEE)
15         theta, phi ← minimizer(cost_function) # some optimizer
16         F ← U(theta, phi) * E  # where U as in Eq. (18)
17         cum_prob ← 0.0
18         r ← rand()  # random threshold for stochastic update
19         for O in F
20             op ← kronecker_pad_with_identity(O, qubit)
21             p ← psi† * op† * op * psi
22             cum_prob ← cum_prob + p
23             if r < cum_prob then
24                 psi ← (1 / sqrt(p)) * op * psi
25                 break
26             endif
27         endfor
28     endfor
29 endfor
30
31 function precompute_overlaps(psi, E)
32     for k, l in 1:dim(E)
33         o[k, l] ← psi† * E[k]† * E[l] * psi  # Eq.(29)
34     endfor
35     return o
36 end
37
38 function non_unitarity_cost(theta, phi, os, trEEEE)
39     u ← U(theta,phi)  # defined in Eq.(18)
40     cost ← 0.0
41     for j in 1:2  # two Kraus operators
42         p ← sum(
43             conj(u[j,k]) * u[j,l] * os[k,l] for k,l in 1:2
44         )  # Eq.(27)
45         tr4F ← sum(
46             conj(u[j,a]) * u[j,b] * conj(u[j,c]) * u[j,d]
47             * trEEEE[a,b,c,d] for a,b,c,d in 1:2
48         )  # Eq.(28)
49         cost ← cost + real(tr4F) / real(p)
50     endfor
51     return −cost
52 end
```

Listing 1: Pseudocode for simulating a noisy quantum circuit using NUMU.

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
