# Peer review of "Non-unitarity maximizing unraveling of open quantum dynamics"

_SciPost Physics, doi:SciPost Phys. 18, 048 (2025)_

## Round 1 · Referee Report · Anonymous (Referee 1) · 2024-12-24

Strengths
-
The paper touches on a very interesting and important problem: Classical simulability of open quantum systems. In light of the measurement-induced phase transition (MPT) it becomes obvious that when unraveling the density matrix in to quantum trajectories, there is a barrier on the coupling to environment above which the bond-dimension needed to store each trajectory in an MPS becomes bounded. This can be exploited to simulate non-trivial quantum dynamics. This paper explores certain issues in this interesting avenue, which was largely overlooked.
-
If correct the results are strong. Namely, in certain regimes (probably close to the MPT) their scheme of optimizing their cost function is extremely successful; it allows to simulate regimes in phase space that naively un-simulable.
-
Numerical techniques are strong. The authors are proficient in heavy numerical simulations and perform a thorough non-trivial computations, which will surely become an important reference.
Weaknesses
I also have some concerns that I raise in the report below.
Report
The paper attacks a very important question in my opinion; taking advantage of the MPT to simulate dynamics of open quantum systems. In particular, they investigate the prospects of enhancing the range of classically simulable problems by taking advantage of a gauge freedom of the Kraus operators representing the different trajectories to reduce the relevant numerical cost of each trajectory. The idea is very neat and interesting. The results seem to be very strong (fig 1b, fig4 and fig6 in particular) - they get access to regions in the parameter space that are naively unreachable. Therefore, I am generally a strong advocate of this work and would like to see it published. I do have a few comments and concerns that I would be grateful if the authors could take under their consideration:
-
I think the quantity defined as "trajectory non-unitarity" is related to a quantity that was already defined in https://arxiv.org/pdf/1803.00089 and named "operator entanglement" (to be distinguished from the one discussed in the current paper). Namely, I believe that the trace(F^+F F^+F) in the second term in Eq. 25 is proportional to the second operator entanglement (see Eq. 115 in PRX QUANTUM 2, 010352 (2021)). The trace is basically a purity of a "wave-function" representation of the Kraus operators, while the Renyi entropies defined in the above reference is a log of that purity. I can also make a connection to the mixed state entropy defined in PRX 10, 041020 (2020) in the limit where the initial density matrix is maximally mixed. Given that these quantities are indeed fundamental indicators of the MPT it makes perfect sense to maximize them to improve the simulation cost. I think it is interesting to make these connections.
-
The whole point of the Haar gates is that their distribution is invariant under unitary transformations. Naively, this implies that the local unitary basis change U in Eq.2 should not make a difference and can be absorbed into the Haar gates constructing the circuit. This will take the specific trajectory with F's to some other trajectory from the same distribution. So the way I understand the computational advantage of this method is that it takes advantage of the fluctuations in the entanglement structure between one trajectory to another. Therefore, if FOR EVERY TRAJECTORY we choose the angles theta and phi to minimize the cost function we can reduce it on average. This implies that U depends on the trajectory. But then I am confused... If this is true, why does the fixed U with theta = pi/4 give a different result compared with what is called "naive" as shown in fig1b and fig4. Where did my logic go wrong?
-
In some sense Haar gates are the worst adversary for this technique (because of the point I raised in 2). It would be very interesting to test this technique on other models like Clifford or an actual physical Hamiltonian. Have the authors considered these directions?
-
I think there is some typo in fig4. Are the labels in the panels switched compared to the caption?
Requested changes
Please address my questions
Recommendation
Publish (surpasses expectations and criteria for this Journal; among top 10%)
Author: Ruben Daraban on 2025-01-13 [id 5112]
(in reply to Report 1 on 2024-12-24)
We would like to thank the referee for his thoughtful review of our paper. We are also thankful for the recommendation of publication and address each of the comments below.
- We thank the referee for pointing out this connection. We added a comment in section 3 of our submission, in the last paragraph on page 10 that points out this fact, referencing Eq. (115) of PRX QUANTUM 2, 010352 (2021).
- We thank the referee for asking this question. In order to resolve it, we need to clarify a key point about the unitary degree of freedom of the Kraus operators, U: It cannot be absorbed into the Haar gates. This is because U acts specifically on an ancilla qubit after it has been entangled with a system qubit. U can be conceptually considered as a change of the measurement basis of the ancilla qubit. In our numerical simulations, U corresponds to a rotation in the set of Kraus operators. Absorbing U into the circuit would require a redefinition of the circuit structure, including introducing additional ancilla qubits, and making random gates act on more than two qubits. We recognize that this may have been confusing in the original manuscript, since the role of U may not be visually apparent in Fig. 1(a). We therefore modified the inset of panel (a), and the caption, to clarify that the entangling gate with the ancilla qubit is not connected to the Haar random gate. We also turned the equation defining the gate entangling the system with the ancilla qubit into a numbered equation [Eq.(19) in the revised manuscript].
- We thank the referee for the suggestion to explore the technique with Clifford gates or physical Hamiltonians. While we have considered these alternatives, we chose to focus on Haar random gates because they provide a clear model for showcasing NUMU in a worst-case scenario. We agree that exploring more structured models could be an interesting direction for future work, and now added this to the first paragraph of the Conclusion & Outlook, section 5.
- We thank the referee for pointing out the typo in the caption of figure 4. We fixed it.
Author: Ruben Daraban on 2025-01-13 [id 5111]
(in reply to Report 2 on 2024-12-29)We would like to thank the referee for his thoughtful review of our paper, as well as for his recommendation of publication. We are pleased to hear that the referee finds the paper well-written, clear, and of interest to the community.
In response to the suggestion regarding the numerical implementation of the maximization of the "averaged post-channel non-unitarity," we have expanded the description of the algorithm in the revised manuscript by adding a specific appendix with a pseudocode implementation of our technique (Appendix F). We hope that the additional details will provide more clarity and address this point.
Regarding the question on the connection between the Schmidt rank of trajectories and that of density matrices, we are not aware of any general mathematical results establishing a formal link or inequalities between these two quantities. If such results exist, they are not known to us. Nonetheless, we believe this is an interesting direction for future exploration and have added a brief note in the discussion to acknowledge this.

---

## Round 1 · Referee Report · Anonymous (Referee 2) · 2024-12-29

Strengths
1- Very thorough analysis of an important problem (entanglement in quantum trajectories) and proposition of an innovative method for dealing with MPS representability of quantum trajectories. 2- Presentation of state-of-the-art numerical simulations 3- Clear and detailed exposition.
Weaknesses
Report
The crucial novelty of the article is a technique called "trajectory non-unitarity maximizing unraveling" (NUMU): the dynamics is unraveled maximizing a quantity dubbed "averaged post-channel non-unitarity". The remarkable result of the authors is that in several regimes where standard unraveling produced quantum trajectories with entanglement entropies blowing exponential in time, it is possible to judiciously find unravelings that have a more limited entanglement content and that are computationally more advantageous. The result is remarkable, as it opens the path for investigation of many-body open quantum systems using advanced numerical techniques.
The article is well written, thorough, and supported by state-of-the-art numerical results. It meets the requirements for publication in SciPost.
Requested changes
I do not have any particular criticism concerning this paper, that I find well written and clear, and that I am sure will interest many colleagues.
I myself would have liked a more extensive discussion of the actual numerical implementation of the maximization of the "averaged post-channel non-unitarity". If the authors could write a bit more than what they do, I think it would be appreciated.
Another point which I find interesting is the one where the authors compare the bond link of quantum trajectories in MPS representation with the bond link of density matrices in MPDO representation (section 4.2). The authors write "This suggests that there is still untapped potential for lowering entanglement epsilon, MPDO in the chosen unraveling of QT+MPS.". What is known mathematically on the link between the Schmidt rank of trajectories and of density matrices? Are the two quantities completely unrelated or are there inequalities between the two? If anything like that has ever been proved, it would be interesting to recall it here.
Recommendation
Publish (easily meets expectations and criteria for this Journal; among top 50%)

---

## Round 2 · Author Response

In response to their comments and remarks, we have modified the manuscript as described in detail below.
Changes include: addressing the feedback provided by the reviewers, fixing typographical errors, and adding references.
We believe that the updates helped to enhance the clarity and completeness of the manuscript.

---

## Round 2 · List of Changes

- Improved Figure 1 and revised its caption in response to a question of Referee 2.
- Converted an inline equation into a numbered equation (Eq. (19), page 9).
- Removed a superfluous 1/2 factor in Eqs. (23) and (24), with no impact on later equations.
- Added a reference in the last paragraph of page 10, suggested by a Referee 2 (reference [53] in v2).
- Corrected a constant error in Eq. (32) (revised from a factor of 2 to the trace of the identity).
- Fixed a caption error in Figure 4, noted by Referee 2.
- Reformulated the last paragraph of Section 4.2, addressing a comment of Referee 1.
- Added a clarifying sentence to the first paragraph of Section 5.
- Included a pseudocode implementation of NUMU in Appendix F, in response to a comment of Referee 1.

---

## Editorial Decision

published